# A Gradient Guidance Perspective on Stepwise Preference Optimization for Diffusion Models

**Joshua Tian Jin Tee**     **Hee Suk Yoon**
**Abu Hanif Muhammad Syarubany**     **Eunseop Yoon**     **Chang D. Yoo**[*]
Korea Advanced Institute of Science and Technology (KAIST)
{joshuateetj, hskyoon, hanif.syarubany, esyoon97, cd_yoo}@kaist.ac.kr

## Abstract

Direct Preference Optimization (DPO) is a key framework for aligning text-to-image models with human preferences, extended by Stepwise Preference Optimization (SPO) to leverage intermediate steps for preference learning, generating more aesthetically pleasing images with significantly less computational cost. While effective, SPO's underlying mechanisms remain underexplored. In light of this, we critically re-examine SPO by formalizing its mechanism as gradient guidance. This new lens shows that SPO uses biased temporal weighting, giving too little weight to later generative steps, and unlike likelihood centric views it reveals substantial noise in the gradient estimates. Leveraging these insights, our GradSPO algorithm introduces a simplified loss and a targeted, variance-informed noise reduction strategy, enhancing training stability. Evaluations on SD 1.5 and SDXL show GradSPO substantially outperforms leading baselines in human preference, yielding images with markedly improved aesthetics and semantic faithfulness, leading to more robust alignment. Code and models are available at https://github.com/JoshuaTTJ/GradSPO.

## 1   Introduction

The rise of diffusion models has transformed the generation of high-quality images from textual prompts, representing a major leap forward in generative artificial intelligence. Traditional text-to-image (T2I) models [1–3] typically rely on a single-stage training process, learning to synthesize images directly from large-scale paired text-image datasets. While this approach has yielded impressive results, it lacks an explicit mechanism to adapt outputs to align with user preferences or specific application needs. In contrast, large language models (LLMs) have embraced a more nuanced multi-stage training strategy. They undergo extensive pretraining on large, diverse, and sometimes noisy datasets, followed by a fine-tuning stage on datasets annotated with human preferences [4, 5]. This fine-tuning step is crucial for enhancing the models' practical utility, safety, and responsiveness, all while preserving their broad foundational knowledge. Applying this two-stage training approach to text-to-image diffusion models presents a promising opportunity to better align generated images with human preferences, resulting in outputs that more faithfully capture user intent.

Recent advances in language modeling have sparked increasing interest in incorporating human preference feedback into the training of text-to-image diffusion models. This emerging area leverages human judgments to refine model outputs, thereby enhancing both alignment with user intent and overall visual fidelity. Prominent works in this field are mostly based on Reinforcement Learning from Human Feedback (RLHF), which uses human comparisons to rank generated images for guiding model training [6–9]. Among these, Diffusion DPO  [6] adapts the Direct Preference

---

[*]Corresponding Author

Optimization (DPO) framework—initially introduced for Large Language Models (LLMs)—to the unique characteristics of diffusion models. This specific adaptation enables the model to inherently learn to favor higher-quality images, and consequently, this strategy has demonstrated notable improvements in producing images that are both visually appealing and semantically aligned with user prompts. Building on this foundation, a recent work, Stepwise Preference Optimization (SPO) [10] introduces preference learning at intermediate diffusion steps, providing more precise reward signals during the denoising process. This approach enhances training efficiency by improving the stepwise likelihood of favorable transitions throughout the diffusion trajectory.

While effective, the SPO framework introduces a learning paradigm that diverges from traditional diffusion training; the latter based on distribution matching, whereas SPO relying on maximizing sample-wise likelihoods. This paper presents a critical re-examination of SPO, positing that its mechanism can be more accurately and beneficially characterized as a form of Direct Preference Optimization (DPO) with respect to score functions augmented by the gradients of reward models. This novel theoretical lens is instrumental: firstly, it reveals an inherent bias in SPO's implicit weighting of generative timesteps, leading to an underemphasis on crucial final-stage details. Secondly, by reframing the optimization from a likelihood-based perspective to a gradient-centric one, our approach uniquely facilitates the quantification and analysis of noise inherent in the preference-guided learning process, specifically through the variance of the guiding reward gradient approximations.

**Our Contributions.** Building upon the re-examination of SPO through a gradient guidance lens, this paper introduces GradSPO and makes the following key contributions:

- **Novel Theoretical Framework for SPO:** We formally reinterpret SPO through the lens of guided score matching. This perspective uncovers an inherent bias in SPO's weighting of generative timesteps and introduces a new method for quantifying learning noise via the variance of gradient approximations.
- **Principled Design of GradSPO:** Building on this framework, we propose GradSPO, an algorithm that incorporates: (1) a simplified loss objective with uniform timestep weighting and fixed guidance scale to directly address the identified SPO from this new viewpoint, and (2) an integrated noise reduction strategy grounded in the gradient perspective to improve training stability and enhance preference fidelity.
- **State-of-the-Art Preference Alignment:** Through extensive experiments, we show that GradSPO significantly outperforms existing preference learning baselines, achieving superior alignment with fine-grained human preferences in text-to-image diffusion models, leading to notable gains in both visual quality and semantic accuracy.

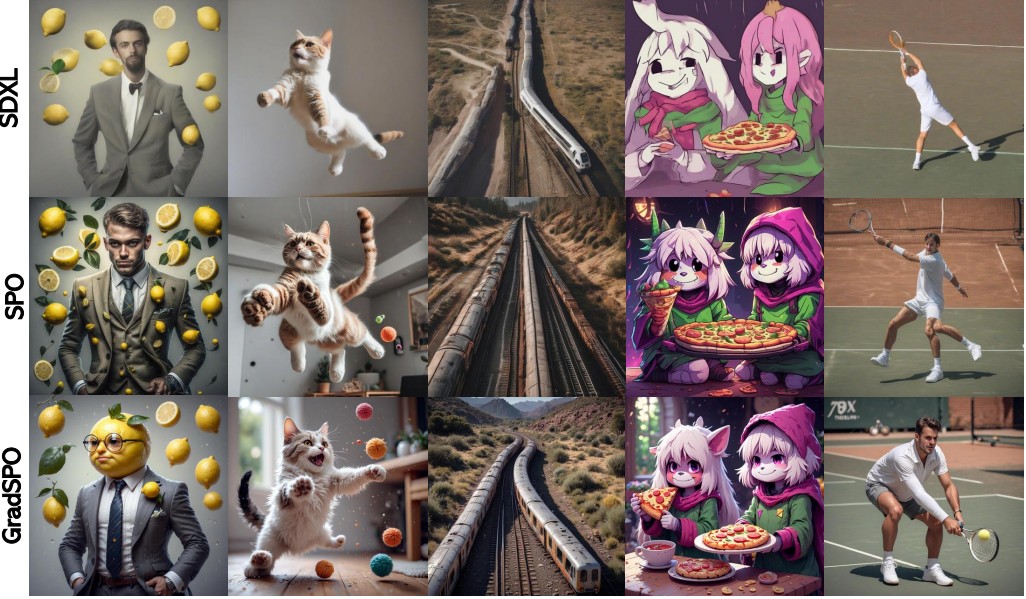

**Figure 1:** Qualitative comparison between vanilla SDXL, SPO, and GradSPO. GradSPO demonstrates superior aesthetic quality and prompt alignment compared to both SDXL and SPO.

## 2 Notations and Preliminaries

This section introduces the foundational concepts and notation for diffusion models and preference optimization techniques relevant to our work.

### 2.1 Diffusion Models

Diffusion models define a forward noising process that gradually transforms a clean data sample $\mathbf{x}_0 \in \mathbb{R}^d$ into pure noise through a sequence of latent variables $\mathbf{x}_1, \ldots, \mathbf{x}_T$, where $T$ denotes the total number of timesteps. Each $\mathbf{x}_t$ represents a progressively noisier version of $\mathbf{x}_0$. The forward process is defined as a Markov chain:

$$q(\mathbf{x}_{1:T}|\mathbf{x}_0) := \prod_{t=1}^{T} q(\mathbf{x}_t|\mathbf{x}_{t-1}), \quad q(\mathbf{x}_t|\mathbf{x}_{t-1}) := \mathcal{N}(\mathbf{x}_t; \sqrt{1 - \beta_t}\mathbf{x}_{t-1}, \beta_t \mathbf{I}), \tag{1}$$

where $\beta_t \in (0, 1)$ controls the noise variance at timestep $t$. A useful property of this process is that $\mathbf{x}_t$ can also be sampled directly from $\mathbf{x}_0$ via a closed-form expression:

$$q(\mathbf{x}_t|\mathbf{x}_0) = \mathcal{N}(\mathbf{x}_t; \sqrt{\bar{\alpha}_t}\mathbf{x}_0, (1 - \bar{\alpha}_t)\mathbf{I}),$$

where $\alpha_t = 1 - \beta_t$ and $\bar{\alpha}_t = \prod_{s=1}^{t} \alpha_s$ is the cumulative product of $\alpha_s$ up to timestep $t$.

The generative model seeks to reverse this process by learning a parameterized reverse distribution $p_\theta(\mathbf{x}_{t-1}|\mathbf{x}_t, \mathbf{c})$, where $\mathbf{c}$ denotes optional conditioning information (e.g., a text prompt). This reverse process is modeled using a neural network $\epsilon_\theta(\mathbf{x}_t, \mathbf{c}, t)$, which predicts the noise component $\epsilon$ that was added to $\mathbf{x}_0$ to obtain $\mathbf{x}_t$. Following Ho et al. [11], each reverse transition is defined as:

$$p_\theta(\mathbf{x}_t|\mathbf{x}_{t+1}, \mathbf{c}) = \mathcal{N}\left(\mathbf{x}_t; \boldsymbol{\mu}_\theta(\mathbf{x}_{t+1}, \mathbf{c}, t+1), \sigma_{t+1}^2 \mathbf{I}\right),$$

$$\text{where} \quad \boldsymbol{\mu}_\theta = \sqrt{\frac{\alpha_t}{\alpha_{t+1}}}\left(\mathbf{x}_{t+1} - \frac{\beta_{t+1}}{\sqrt{1 - \bar{\alpha}_{t+1}}}\epsilon_\theta(\mathbf{x}_{t+1}, \mathbf{c}, t+1)\right), \quad \sigma_{t+1}^2 = \frac{1 - \bar{\alpha}_t}{1 - \bar{\alpha}_{t+1}}\beta_{t+1}. \tag{2}$$

The model is trained by minimizing the variational bound, which reduces to a simplified objective known as the denoising score matching loss:

$$\mathcal{L}_{\text{DDPM}} = \mathbb{E}_{\mathbf{x}_0, \mathbf{c}, t, \epsilon \sim \mathcal{N}(0, \mathbf{I})}\left[\lambda(t)\left\|\epsilon - \epsilon_\theta(\sqrt{\bar{\alpha}_t}\mathbf{x}_0 + \sqrt{1 - \bar{\alpha}_t}\epsilon, \mathbf{c}, t)\right\|^2\right], \tag{3}$$

where $t$ is sampled uniformly from $\{1, \ldots, T\}$, and $\lambda(t)$ is a time-dependent weighting function. The term inside the expectation represents the squared error between the true noise $\epsilon$ and the model's prediction $\epsilon_\theta$ at timestep $t$.

### 2.2 Stepwise Preference Optimization (SPO)

Among various preference alignment training schemes, many recent methods adopt a DPO-style training framework [12, 6, 8, 10, 13]. Given a preference dataset $\mathcal{D}$, these methods often aim to minimize an objective related to the DPO loss. For Diffusion DPO [6], this can be framed as minimizing an upper bound on the DPO loss [12], defined as:

$$\begin{aligned}
\mathcal{L}_{\text{DiffDPO}} = -\mathbb{E}\bigg[\log \sigma\bigg(&-\beta T\big(\mathbb{D}_{\text{KL}}\left(q(\mathbf{x}_{t-1}^w \mid \mathbf{x}_t^w, \mathbf{x}_0^w, \mathbf{c}) \parallel p_\theta(\mathbf{x}_{t-1}^w \mid \mathbf{x}_t^w, \mathbf{c})\right) \\
&- \mathbb{D}_{\text{KL}}\left(q(\mathbf{x}_{t-1}^w \mid \mathbf{x}_t^w, \mathbf{x}_0^w, \mathbf{c}) \parallel p_{\text{ref}}(\mathbf{x}_{t-1}^w \mid \mathbf{x}_t^w, \mathbf{c})\right) \\
&- \mathbb{D}_{\text{KL}}\left(q(\mathbf{x}_{t-1}^l \mid \mathbf{x}_t^l, \mathbf{x}_0^l, \mathbf{c}) \parallel p_\theta(\mathbf{x}_{t-1}^l \mid \mathbf{x}_t^l, \mathbf{c})\right) \\
&+ \mathbb{D}_{\text{KL}}\left(q(\mathbf{x}_{t-1}^l \mid \mathbf{x}_t^l, \mathbf{x}_0^l, \mathbf{c}) \parallel p_{\text{ref}}(\mathbf{x}_{t-1}^l \mid \mathbf{x}_t^l, \mathbf{c}))\right)\bigg)\bigg],
\end{aligned} \tag{4}$$

where $(\mathbf{x}_0^w, \mathbf{x}_0^l, \mathbf{c}) \sim \mathcal{D}$, $t \sim \mathcal{U}[0, T]$, and samples $\mathbf{x}_t^w$ and $\mathbf{x}_t^l$ are drawn from $q(\mathbf{x}_t^w \mid \mathbf{x}_0^w)$ and $q(\mathbf{x}_t^l \mid \mathbf{x}_0^l)$ respectively.

While the original Diffusion-DPO [6] formulation often utilizes clean preference pairs $(\mathbf{x}_0^w, \mathbf{x}_0^l)$ from offline datasets to approximate such objectives, the recently introduced Stepwise Preference Optimization (SPO) [10] approach leverages intermediate samples generated by an online model as preference pairs. The SPO loss is typically expressed in a more direct log-likelihood ratio form, analogous to the general DPO principle:

$$\mathcal{L}_{\text{SPO}} = -\mathbb{E}_{\mathbf{c}, \mathbf{x}_t, t} \left[ \log \sigma \left( \beta \left( \log \frac{p_\theta(\mathbf{x}_{t-1}^w \mid \mathbf{c}, \mathbf{x}_t)}{p_{\text{ref}}(\mathbf{x}_{t-1}^w \mid \mathbf{c}, \mathbf{x}_t)} - \log \frac{p_\theta(\mathbf{x}_{t-1}^l \mid \mathbf{c}, \mathbf{x}_t)}{p_{\text{ref}}(\mathbf{x}_{t-1}^l \mid \mathbf{c}, \mathbf{x}_t)} \right) \right) \right]. \tag{5}$$

The timestep $t$ is uniformly sampled as $t \sim \mathcal{U}[1, T - \kappa]$ for a fixed constant $\kappa$. Textual prompts $\mathbf{c}$ are drawn from a distribution $p(\mathbf{c})$, and the initial latent state $\mathbf{x}_T$ is sampled from $\mathcal{N}(\mathbf{0}, \mathbf{I})$. Intermediate latent states $\mathbf{x}_t$ are obtained by applying the discrete reverse diffusion process (using Eq. 2 iteratively). The winning $\mathbf{x}_{t-1}^w$ and losing $\mathbf{x}_{t-1}^l$ samples for the transition from $\mathbf{x}_t$ to $\mathbf{x}_{t-1}$ are determined by a step-aware reward model $r(\cdot, \cdot)$:

$$\mathbf{x}_{t-1}^w = \underset{\mathbf{x}_{t-1} \in \{\mathbf{x}_{t-1}^{(i)}\}}{\operatorname{argmax}} r(\mathbf{x}_{t-1}, t-1), \qquad \mathbf{x}_{t-1}^l = \underset{\mathbf{x}_{t-1} \in \{\mathbf{x}_{t-1}^{(i)}\}}{\operatorname{argmin}} r(\mathbf{x}_{t-1}, t-1), \tag{6}$$

where $\{\mathbf{x}_{t-1}^{(i)}\}$ are candidate samples generated from $p_\theta(\mathbf{x}_{t-1} \mid \mathbf{c}, \mathbf{x}_t)$.

SPO contrasts with standard DPO applications by operating on preferences over intermediate transitions rather than only final outputs. By integrating feedback throughout the generation process, SPO provides denser reward signals. This stepwise learning mechanism aims to enhance training efficiency and offer finer-grained control for aligning the model's generation trajectory with human preferences while requiring significantly less computational cost.

## 3 GradSPO: A Gradient Guidance Perspective on Stepwise Preference Optimization for Diffusion Models

Stepwise Preference Optimization (SPO) [10], as outlined in Section 2.2, aligns diffusion models by optimizing preferences at intermediate generative steps. Recall from Eq. 2 that the generation of $\mathbf{x}_{t-1}$ from a given $\mathbf{x}_t$ and context $\mathbf{c}$ involves sampling noise $z \sim \mathcal{N}(0, \mathbf{I})$ such that $\mathbf{x}_{t-1} = \mu_\theta(\mathbf{x}_t, \mathbf{c}, t) + \sigma_t z$. Consequently, selecting $\mathbf{x}_{t-1}$ to maximize or minimize a step-aware reward $r(\mathbf{x}_{t-1}, t-1)$ (Eq. 6) is equivalent to finding the optimal noise $z$ for that step. This can be expressed as:

$$\begin{aligned} z^+ &= \underset{z}{\operatorname{argmax}} \, r(\mu_\theta(\mathbf{x}_t, \mathbf{c}, t) + \sigma_t z, t-1), \\ z^- &= \underset{z}{\operatorname{argmin}} \, r(\mu_\theta(\mathbf{x}_t, \mathbf{c}, t) + \sigma_t z, t-1). \end{aligned} \tag{7}$$

A key insight from Huang et al. [14] is the interpretation this argmax noise serves as an approximation of a scaled, noisy gradient of the reward model:

$$z^\pm \approx \pm \sqrt{T \beta_t} \nabla_{x_t} r(x_t) + z, \tag{8}$$

where $z \sim \mathcal{N}(0, I)$. This approximation motivates us to view SPO from a reward guided perspective. Within this view, an ideal reward-guided score function can be defined as $s_{\text{ideal}}^{w,l}(\mathbf{x}_t, \mathbf{c}, t) = s_\theta(\mathbf{x}_t, \mathbf{c}, t) \pm \gamma \nabla_{\mathbf{x}_t} r(\mathbf{x}_t, t)$, where $s_\theta(\mathbf{x}_t, \mathbf{c}, t) = \nabla_{\mathbf{x}_t} \log p_\theta(\mathbf{x}_t | \mathbf{c})$ is the original model score and $\gamma$ is the guidance scale. From this point of view, SPO can be seen as approximating this ideal guidance term $\pm \gamma \nabla_{\mathbf{x}_t} r(\mathbf{x}_t, t, t)$ relying on the connection between the gradient of the reward models and $z^\pm$ in Eq. 8. Specifically, we define our approximate guided scores $\hat{s}_\theta^{w,l}$ as:

$$\hat{s}_\theta^{w,l}(\mathbf{x}_t, \mathbf{c}, t) := \nabla_{\mathbf{x}_t} \log p_\theta(\mathbf{x}_t | \mathbf{c}) + \gamma_t' z^\pm \tag{9}$$

Here, $z^\pm$ is defined in Eq. 7, and $\gamma_t'$ is a time-dependent scaling factor related to $\gamma$. Drawing an analogy to how Diffusion DPO handles preferred and rejected samples, we introduce GradSPO, a

method that performs model alignment by extending DPO principles to operate on guided scores:

$$
\begin{aligned}
L_{\text{GradSPO}}(\theta, \theta_{\text{ref}}; \mathbf{x}_t, \mathbf{c}, t) = -\mathbb{E}\bigg[ \log \sigma \bigg( &-\beta w(t) \big( \| s_\theta(\mathbf{x}_t, \mathbf{c}, t) - \text{sg}(\hat{s}_\theta^w(\mathbf{x}_t, \mathbf{c}, t)) \|_2^2 \\
&- \| s_{\text{ref}}(\mathbf{x}_t, \mathbf{c}, t) - \text{sg}(\hat{s}_\theta^w(\mathbf{x}_t, \mathbf{c}, t)) \|_2^2 \\
&- \| s_\theta(\mathbf{x}_t, \mathbf{c}, t) - \text{sg}(\hat{s}_\theta^l(\mathbf{x}_t, \mathbf{c}, t)) \|_2^2 \\
&+ \| s_{\text{ref}}(\mathbf{x}_t, \mathbf{c}, t) - \text{sg}(\hat{s}_\theta^l(\mathbf{x}_t, \mathbf{c}, t)) \|_2^2 \big) \bigg) \bigg],
\end{aligned}
\tag{10}
$$

where $s_\theta(\mathbf{x}_t, \mathbf{c}, t)$ is the current model's score prediction (e.g., related to $\epsilon_\theta$), $s_{\text{ref}}$ is the reference model's score, $\text{sg}(\cdot)$ denotes the stop-gradient operator (which prevents gradients from flowing through its argument), and $w(t)$ is a time-dependent weighting function. The stop-gradient operator is essential for stable training, as it detaches the target scores, i.e., $\text{sg}(s_{\theta_0}^w)$ and $\text{sg}(s_{\theta_0}^l)$, from gradient updates. With these fixed targets, Eq. 10 trains the model by pulling the current score $s_\theta(\mathbf{x}_t, \mathbf{c}, t)$ closer to the winning score $s_\theta^w$, while pushing it away from the losing score $s_\theta^l$. Without this detachment, the winning score would drift toward the current prediction ($s_\theta^w \to s_\theta$), weakening the supervision signal and destabilizing learning.

**Theorem 1** (GradSPO Loss as Upper Bound). *Let $L_{GradSPO}(s_\theta, s_{ref}; T^w, T^l)$ denote the GradSPO loss functional as defined in Eq. 10, where $T^w$ and $T^l$ are the target winning and losing scores. Let $s_{ideal}^w = s_\theta + \gamma \nabla_{\mathbf{x}_t} r(\mathbf{x}_t, t)$ and $s_{ideal}^l = s_\theta - \gamma \nabla_{\mathbf{x}_t} r(\mathbf{x}_t, t)$ be the ideal target scores based on true reward gradients. Let $\hat{s}_\theta^w$ and $\hat{s}_\theta^l$ (as defined in Eq. 9) be the target scores constructed using the approximated, potentially noisy, gradient signal. Then, the GradSPO loss computed with the approximated targets forms an upper bound on the loss computed with ideal targets:*

$$
L_{GradSPO}(s_\theta, s_{ref}; s_{ideal}^w, s_{ideal}^l) \leq L_{GradSPO}(s_\theta, s_{ref}; \hat{s}_\theta^w, \hat{s}_\theta^l).
\tag{11}
$$

A detailed derivation of Theorem 1 is provided in Appendix A. This theorem is significant as it demonstrates that our practical GradSPO loss (the right-hand side, which we minimize) serves as an upper bound on an idealized loss formulated with exact reward gradients. Analogous to ELBO maximization in variational inference, minimizing this upper bound provides a principled approach to optimizing for the underlying clean-gradient objective. This is particularly advantageous in scenarios like latent diffusion models, where precise gradient computations $\nabla_{\mathbf{x}_t} r(\mathbf{x}_t, t)$ might involve costly backpropagation through components like the VAE decoder. The GradSPO loss objective from Eq. 10, when expressed in terms of noise predictions $\epsilon_\theta(\mathbf{x}_t, \mathbf{c}, t)$ (where $s_\theta(\mathbf{x}_t, \mathbf{c}, t) = -\epsilon_\theta(\mathbf{x}_t, \mathbf{c}, t)/\sqrt{1 - \bar{\alpha}_t}$ can be written as:

$$
\begin{aligned}
\mathcal{L}_{\text{GradSPO}} = -\mathbb{E}\bigg[ \log \sigma \bigg( &-\beta a(t) \big( \| \epsilon_\theta(\mathbf{x}_t, \mathbf{c}, t) - \text{sg}(\epsilon_\theta(\mathbf{x}_t, \mathbf{c}, t) - \Delta\epsilon^+) \|_2^2 \\
&- \| \epsilon_{\text{ref}}(\mathbf{x}_t, \mathbf{c}, t) - \text{sg}(\epsilon_\theta(\mathbf{x}_t, \mathbf{c}, t) - \Delta\epsilon^+) \|_2^2 \\
&- \| \epsilon_\theta(\mathbf{x}_t, \mathbf{c}, t) - \text{sg}(\epsilon_\theta(\mathbf{x}_t, \mathbf{c}, t) - \Delta\epsilon^-) \|_2^2 \\
&+ \| \epsilon_{\text{ref}}(\mathbf{x}_t, \mathbf{c}, t) - \text{sg}(\epsilon_\theta(\mathbf{x}_t, \mathbf{c}, t) - \Delta\epsilon^-) \|_2^2 \big) \bigg) \bigg],
\end{aligned}
\tag{12}
$$

where $\Delta\epsilon^\pm = \gamma_t \sqrt{1 - \bar{\alpha}_t} z^\pm$, $a(t)$ is a time-dependent weighting function, and $\beta$ is a scalar coefficient. Notably, the standard SPO loss can be interpreted within this guided noise prediction framework by specific choices of $a(t)$ and $\gamma_t$ (see Appendix A for details).

This reinterpretation of SPO through a gradient guidance perspective offers critical insights into its behavior and limitations. For instance, it reveals that SPO's interval-based training [10] can be viewed as an application of the recently introduced interval guidance [15], which has been shown to improve sample quality and diversity.

More critically, this re-framing reveals that the effective weighting function $a(t)$ in the original SPO formulation (as shown in Figure 2) disproportionately underweights the later diffusion steps. A similar pattern emerges when analyzing the Pick Score difference between the base model and its SPO-finetuned counterpart: the improvement decays exponentially over time, even though later steps are crucial for capturing fine-grained details. This suggests that SPO primarily enhances early timesteps, with minimal gains in the later stages of the denoising process. To address this imbalance and ensure more uniform supervision across timesteps, we adopt a simplified objective in which both $a(t)$ and $\gamma_t$ are set to constant values.

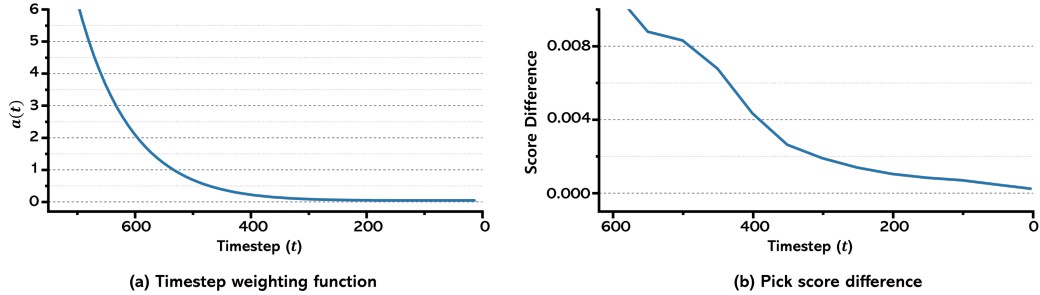

(a) Timestep weighting function          (b) Pick score difference

**Figure 2:** (a) Timestep weighting function $a(t)$ used in SPO. (b) Pick Score difference between SPO and the base model across timesteps.

### 3.1 Noise Reduction Techniques

The $z^{\pm}$ values (Eq. 8), which serve as noisy reward gradient approximations also include an inherent noise component $z \sim \mathcal{N}(0, \mathbf{I})$. This inherent noise in $z^{\pm}$ can introduce variance into the target signals, potentially hindering training performance. To address this, we propose a technique to reduce noise in the estimated gradient direction. Recall $z^{\pm} \approx z \pm \sqrt{T\beta_t}\nabla_{\mathbf{x}_t} r(\mathbf{x}_t, t)$. The term $\pm\sqrt{T\beta_t}\nabla_{\mathbf{x}_t} r(\mathbf{x}_t, t)$ is the desired gradient signal. We define a noise-reduced estimate as:

$$z^{\pm}_{\text{max-min}} = \pm\frac{z^+ - z^-}{2} \tag{13}$$

This estimator leverages the opposing nature of the target gradient components in $z^+$ and $z^-$.

**Theorem 2** (Variance Reduction of Estimated Gradient Signal). *Let $z^+ = G + z_1$ and $z^- = -G + z_2$, where $G = \sqrt{T\beta_t}\nabla_{\mathbf{x}_t} r(\mathbf{x}_t, t)$ is the true scaled gradient signal, and $z_1, z_2 \sim \mathcal{N}(0, \mathbf{I})$ are independent noise terms. The noise-reduced estimate $z^{\pm}_{\text{max-min}}$ has variance half that of the original estimation $z^{\pm}$:*

$$Var(z^{\pm}_{\text{max-min}}) = \frac{Var(z^{\pm})}{2} \tag{14}$$

A proof is provided in Appendix B. This demonstrates that $z^{\pm}_{\text{max-min}}$ offers a more stable estimate of the desired gradient signal, leading to a more stable training process that converges effectively, resulting in the generation of better aligned images.

While noise reduction techniques are beneficial, noise cannot be entirely removed. The persistence of this inherent noise in the gradient estimates necessitates the development of training algorithms that are robust to its presence. Towards this end, to further enhance stability and promote convergence, we employ an Exponential Moving Average (EMA) of the model parameters $\theta$ during training and online sampling. The EMA parameters $\theta_{\text{EMA}}$ are updated as:

$$\theta_{\text{EMA}} \leftarrow \mu\theta_{\text{EMA}} + (1 - \mu)\theta, \tag{15}$$

where $\mu \in [0, 1)$ is a decay rate (e.g., 0.999). EMA parameters often provide a more stable representation of the learned model particularly in noisy environments [16, 17].

By incorporating these noise reduction strategies, we arrive at the final GradSPO framework. The target noise predictions in the loss function are now constructed using the EMA model's prediction $\epsilon_{\theta_{\text{EMA}}}$ and the noise-reduced gradient signal $z^{\pm}_{\text{max-min}}$. The resulting GradSPO loss is:

$$
\begin{aligned}
\mathcal{L}_{\text{GradSPO}} = -\mathbb{E}\Big[\log \sigma\Big(&-\beta a(t)\big(\|\epsilon_\theta(\mathbf{x}_t, \mathbf{c}, t) - \text{sg}(\epsilon_{\theta_{\text{EMA}}}(\mathbf{x}_t, \mathbf{c}, t) - \Delta\epsilon^+_{\text{reduced}})\|_2^2 \\
&-\|\epsilon_{\text{ref}}(\mathbf{x}_t, \mathbf{c}, t) - \text{sg}(\epsilon_{\theta_{\text{EMA}}}(\mathbf{x}_t, \mathbf{c}, t) - \Delta\epsilon^+_{\text{reduced}})\|_2^2 \\
&-\|\epsilon_\theta(\mathbf{x}_t, \mathbf{c}, t) - \text{sg}(\epsilon_{\theta_{\text{EMA}}}(\mathbf{x}_t, \mathbf{c}, t) - \Delta\epsilon^-_{\text{reduced}})\|_2^2 \\
&+\|\epsilon_{\text{ref}}(\mathbf{x}_t, \mathbf{c}, t) - \text{sg}(\epsilon_{\theta_{\text{EMA}}}(\mathbf{x}_t, \mathbf{c}, t) - \Delta\epsilon^-_{\text{reduced}})\|_2^2\big)\Big)\Big],
\end{aligned}
\tag{16}
$$

where $\Delta\epsilon^{\pm}_{\text{reduced}} = \gamma_t\sqrt{1 - \bar{\alpha}_t}z^{\pm}_{\text{max-min}}$. This changes brought about by this new viewpoint on SPO substantially improves the alignment capabilities of stepwise preference optimization.

# 4 Experimental Results

## 4.1 Experimental Setup

**Datasets and Models**. We fine-tune both Stable Diffusion 1.5 [18] (Creativeml-openrail-m License) and SDXL [19] (Openrail++ License) models using the GradSPO objective, as detailed in Section 3. Following the SPO training scheme, we train the models on 4,000 randomly sampled prompts from the Pick-a-Pic v1 dataset [20], which contains 580,000 pairs of image preference for each prompt. For evaluation, unless stated otherwise, we used the test set consisting of 500 prompts sourced from the Pick-a-Pic v2 dataset, similar to previous work in the field [12, 10].

**Implementation Details**. Since GradSPO builds upon the SPO framework [10], we retain SPO's base hyperparameters to ensure a fair and direct comparison. Additionally, because we do not modify the stepwise-aware preference model, we reuse the same reward model from SPO rather than training a new one. However, GradSPO introduces several unique hyperparameters to optimize performance: the time-dependent weight function $\alpha_t$ is set to 1, the guidance scale $\gamma_t$ is fixed at 0.5, and the Exponential Moving Average (EMA) decay rate $\mu$ is set to 0.9. These settings are applied consistently across both SDXL [19] and Stable Diffusion 1.5 [18] models. A complete listing of all hyperparameters used in our experiments can be found in Appendix D for reproducibility and further reference.

**Baselines**. To evaluate the effectiveness of GradSPO in aligning with human preferences, we benchmark its performance against several strong baselines. These include the original pre-trained models, Stable Diffusion 1.5 [18] and SDXL [19], as well as several recent preference learning methods: Direct Preference Optimization (DPO) [12], Stepwise Preference Optimization (SPO) [10], InPO [8], and MaPO [13]. For consistency across comparisons, we utilize publicly available pretrained checkpoints for all baseline methods and apply identical evaluation protocols.

**Evaluation**. We assess model performance using four widely accepted metrics that quantify alignment with human preferences: HPS v2 [21] (Apache-2.0 License), PickScore [22] (MIT License), Aesthetic score [23] (MIT License), and Image Reward [24] (Apache-2.0 License). For each prompt in the evaluation set, we compute and report the average scores across all compared models to provide a comprehensive view of performance. In line with the evaluation procedure used in SPO [10], we generate images for all models using DDIM sampling [25] with 20 diffusion steps and a classifier-free guidance scale of 5.0 [26]. This consistent inference setup ensures fair comparisons of image quality and alignment metrics across methods.

## 4.2 Quantitative Results

Table 1 presents quantitative comparisons of GradSPO against established baselines on both SD 1.5 and SDXL backbones. For the SD 1.5 backbone, GradSPO demonstrates strong performance, surpassing most existing methods. While InPO achieves a marginally higher score on the Image Reward metric, it is important to contextualize this: GradSPO builds upon the SPO framework, which itself registers the lowest Image Reward among the compared alignment techniques. Despite this foundational starting point, GradSPO substantially elevates SPO's Image Reward score to 0.4747, securing the second-highest position for this metric. Furthermore, GradSPO markedly improves SPO's aesthetic score, underscoring the significant advantages of reinterpreting SPO through our gradient guidance perspective.

On the SDXL backbone, GradSPO's performance is particularly compelling, demonstrating clear superiority across all evaluated metrics. It achieves, for instance, a leading Aesthetic Score of 6.2985 and a Pick Score of 28.93. The improvement over its direct precursor, Stepwise Preference Optimization (SPO), is significant—boosting the HPSv2 score from 28.27 (SPO) to 28.93 (GradSPO). These outcomes strongly affirm the benefits of our gradient guidance reinterpretation of SPO, particularly its advantages over conventional likelihood-based optimization strategies.

To further assess the effectiveness of our GradSPO training paradigm on SDXL, we conducted a user study with five judges using 100 prompts randomly sampled from the HPSv2 benchmark [21]. For each prompt, participants were shown two images, one generated by GradSPO and the other by a competing method, and asked to indicate a preference or select a tie if neither image was clearly better. Judgments were based on three criteria: overall image quality, image–text alignment, and aesthetic appeal.

| Model | Method | HPSv2 | Pick Score | Aesthetic Score | Image Reward |
|-------|--------|-------|------------|-----------------|--------------|
| SD 1.5 [18] | Baseline | 26.26 | 20.62 | 5.2687 | 0.0741 |
| | DPO [6] | 26.56 | 21.01 | 5.3704 | 0.2704 |
| | InPO [8] | **26.86** | 21.21 | 5.4674 | **0.5135** |
| | SPO [10] | 26.47 | 21.11 | 5.5898 | 0.1945 |
| | **GradSPO (Ours)** | **26.86** | **21.38** | **5.7651** | 0.4747 |
| SDXL [19] | Baseline | 27.06 | 21.85 | 5.8253 | 0.4749 |
| | DPO [6] | 27.81 | 22.41 | 5.8412 | 0.7466 |
| | MaPO [13] | 27.30 | 21.95 | 5.9684 | 0.5868 |
| | InPO [8] | 28.07 | 22.46 | 5.9046 | 0.8546 |
| | SPO [10] | 28.27 | 22.93 | 6.2236 | 0.9982 |
| | **GradSPO (Ours)** | **28.93** | **23.45** | **6.2985** | **1.0861** |

Table 1: **Comparison of GradSPO with baseline methods on SD 1.5 and SDXL backbones.** GradSPO attains the highest scores across most human preference metrics, demonstrating superior alignment and visual quality. For each metric, the top-performing method is **bolded**, while the second-best is underlined.

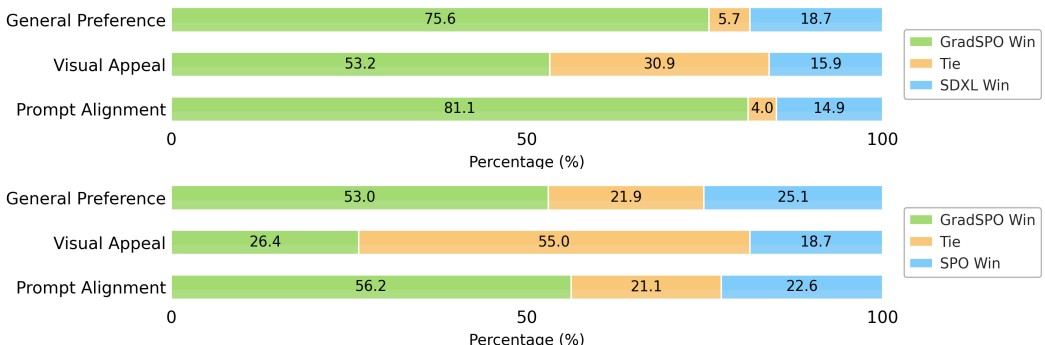

**Figure 3:** User study results comparing GradSPO against two baselines. Top: comparison with SDXL. Bottom: comparison with SPO.

As summarized in Figure 3, the results indicate a clear human preference for images generated by GradSPO, reflecting a notable improvement in overall quality. GradSPO achieved a win rate of 75.6% against SDXL and 53.0% against SPO. Notably, the strongest gains were observed in aesthetic quality, where GradSPO attained win rates of 81.1% and 56.2% against SDXL and SPO, respectively.

## 4.3 Qualitative Results

Figure 4 illustrates the qualitative performance of our model, GradSPO (using the SDXL backbone), compared to other Text-to-Image (T2I) methods. In the first row, for the prompt "The Best," GradSPO successfully generates an image accurately containing the desired text, outperforming baseline methods. Notably, while the InPO approach nearly generates the correct text, it includes visible artifacts. For the second row, corresponding to the prompt "Buff Harry Potter," most baseline methods (excluding InPO) fail to generate images following the provided textual prompt. However, comparing GradSPO and InPO, GradSPO produces a more aesthetically pleasing image. Lastly, in the final row, all baseline methods fail to effectively render the "A gorgeous queen" following the textual prompt, whereas GradSPO reliably and accurately generates the image. Collectively, these examples in Figure 4 qualitatively demonstrate GradSPO's superior capability in prompt adherence, text rendering, and aesthetic quality compared to baseline methods.

## 4.4 Ablation on EMA Momentum and Reward Guidance Scale

To better understand the performance and stability of GradSPO, we conduct ablations on two key hyperparameters: the EMA momentum $\mu$ and the reward guidance scale $\gamma$. All other settings are fixed for SD 1.5, with results reported on the Pick-a-Pic v2 dataset. When varying $\mu$, we set $\gamma = 0.5$, and when varying $\gamma$, we fix $\mu = 0.9$. The results are summarized in Table 2.

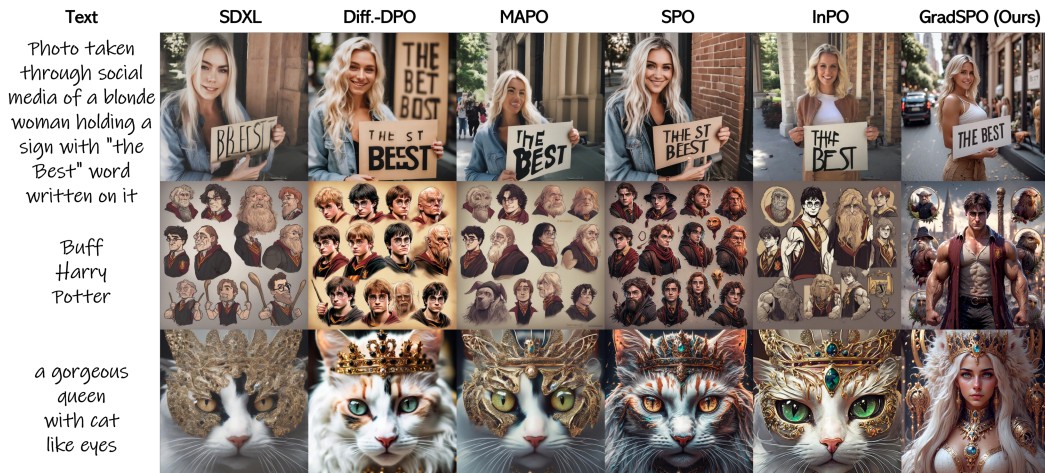

| Text | SDXL | Diff.-DPO | MAPO | SPO | InPO | GradSPO (Ours) |

**Figure 4: Side-by-side comparison of images generated by related methods using SDXL.** GradSPO demonstrates a significant improvement in terms of aesthetic appeal and fidelity to the caption.

| $\gamma$ | HSPv2 | Pick Score | Aesthetic | ImageRwd | | $\mu$ | HSPv2 | Pick Score | Aesthetic | ImageRwd |
|---|---|---|---|---|---|---|---|---|---|---|
| 0.25 | 26.67 | 20.71 | 5.5278 | 0.452 | | 0.90 | **26.86** | **21.38** | **5.7651** | **0.4747** |
| 0.50 | **26.86** | **21.38** | **5.7651** | **0.4747** | | 0.95 | 26.74 | 21.12 | 5.6352 | 0.4153 |
| 0.75 | 26.70 | 20.99 | 5.4630 | 0.3878 | | 0.99 | 26.68 | 21.22 | 5.7591 | 0.3871 |
| | | **(a)** | | | | | | **(b)** | | |

**Table 2:** Ablation study of GradSPO on SD 1.5 (Pick-a-Pic v2). (a) Varying the reward guidance scale $\gamma$ with EMA momentum fixed at $\mu = 0.9$; (b) Varying the EMA momentum $\mu$ with reward scale fixed at $\gamma = 0.5$.

We observe that GradSPO is relatively robust to changes in the EMA momentum $\mu$, showing only minor performance degradation at higher values. We attribute this to an oversmoothing effect, where the EMA model lags behind the current model and adapts too slowly.

In contrast, the reward guidance scale $\gamma$ exerts a stronger influence. A large $\gamma$ increases the variance of the reward signal, especially in noisy regions, leading to less stable convergence. Conversely, a very small $\gamma$ weakens the learning signal, preventing the model from capturing meaningful preferences. Empirically, we find $\gamma = 0.5$ provides a good balance between stability and signal strength, and adopt this setting across both SDXL and SD 1.5 experiments.

## 4.5 Effectiveness of Gradient-Guided Objectives and Noise Mitigation Strategies

To dissect the contributions of our gradient guidance perspective on Stepwise Preference Optimization (SPO) and to validate the efficacy of the proposed noise mitigation strategies, we conducted systematic ablation experiments. The results for experiments conducted on the SDXL backbone, presented in Figure 5, illustrate the progressive performance enhancements (measured by win-rates across diverse automated reward models) as each component of GradSPO is incrementally introduced, starting from a standard SPO baseline.

First, adopting the simplified objective derived from our gradient-guided reinterpretation (denoted SPO+Simple) led to improvements across most metrics, with only a slight drop in aesthetic score. This highlights the advantage of our reformulated objective, including uniform timestep weighting.

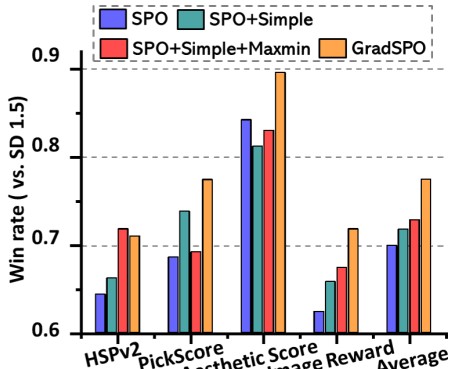

**Figure 5:** Win-rate comparisons across SPO-Simple, SPO-Simple-Maxmin ($z^{\pm}_{\text{max-min}}$), and GradSPO on popular reward metrics,

Subsequently, integrating the max-min noise reduction technique for the gradient signal, $z^{\pm}_{\text{max-min}}$ (termed SPO+Simple+Maxmin), yielded a further discernible boost in performance, increasing the

average win rate across all 4 metrics. This gain is primarily attributed to the reduced variance in the estimated reward gradients, demonstrating the effectiveness of this stabilization approach.

Finally, the incorporation of Exponential Moving Average (EMA) for model parameters during training (Eq. 15), which constitutes the full GradSPO method, achieved the highest average performance among all configurations. This result highlights EMA's crucial role in further stabilizing the training dynamics within the inherently stochastic preference learning landscape.

In summary, the ablation study (Figure 5) demonstrates that each core component of GradSPO—the simplified gradient-guided objective, max-min noise reduction, and EMA—makes a distinct and measurable contribution to the overall performance

## 5 Related Works

### 5.1 Diffusion Models for Text-to-Image Generation

Denoising Diffusion Probabilistic Models (DDPMs) [11] are a class of powerful generative models that have recently gained popularity for their ability to effectively model high-dimensional data. This has led to their successful adoption across a range of applications, including image synthesis [2, 1, 19], video generation [27–29], and text-to-speech synthesis [30–32]. Among these, text-to-image generation has been particularly impactful, enabling the creation of complex visuals directly from textual descriptions [2, 3] and unlocking new possibilities in creative fields such as digital art and design. Despite their ability to produce high-quality images after pretraining, these models often struggle to capture nuanced human preferences, frequently resulting in visual artifacts such as poorly rendered hands and faces [7]. These limitations have spurred ongoing research into improving sampling efficiency [33–35] and enhancing alignment with textual inputs [6, 10, 8].

### 5.2 Human-Preference Alignment for Diffusion

Human preference alignment has long been recognized as beneficial in Large Language Models (LLMs), where techniques like Reinforcement Learning from Human Feedback (RLHF) have substantially improved performance, helpfulness, and safety [5, 4, 12, 36, 37]. Inspired by these successes, recent work has explored applying human preference learning to text-to-image generation [6, 7]. A prominent example is Diffusion DPO [6], which adapts Direct Preference Optimization (DPO) for diffusion models, steering the model toward preferred images and away from dispreferred ones using an offline dataset. While Diffusion DPO significantly enhances text alignment and aesthetic quality, generating high-quality images, it demands considerable computational resources. More recently, Stepwise Preference Optimization (SPO) [10] was introduced, performing preference learning at intermediate diffusion steps on a per-sample basis. This approach provides stronger learning signals, greatly improving computational efficiency and text alignment.

Despite SPO's success, its sample-wise formulation deviates markedly from the typical score matching training paradigm used in diffusion models [11]. In this work, we reinterpret SPO through a score matching lens, showing that rather than maximizing sample-wise likelihoods, SPO effectively learns to move towards preferred scores and away from dispreferred ones. This perspective exposes limitations in the original SPO training framework. Building on this insight, we propose GradSPO, a score matching–inspired approach that substantially enhances image generation quality.

## 6 Conclusion

In this paper, we revisit Stepwise Preference Optimization (SPO) through the lens of score matching. We establish a novel theoretical connection, demonstrating that SPO is equivalent to Direct Preference Optimization (DPO) when using winning and losing score functions derived from reward models with added noise. Leveraging this perspective, we propose a simplified and more intuitive optimization objective, alongside effective noise reduction techniques that significantly mitigate approximation errors caused by such noise. Empirical evaluations demonstrate that our proposed method, GradSPO, consistently outperforms existing preference learning approaches, highlighting its superior capability in generating images that align closely with human preferences. Furthermore, this new interpretation of SPO links gradient guidance to the SPO training objective allowing for the integration of improved gradient guidance techniques for more user aligned images which we leave for future work.

# 7 Acknowledgements

This work was supported by Institute for Information & communications Technology Planning & Evaluation (IITP) grant funded by the Korea government(MSIT) (No.RS-2021- II211381, Development of Causal AI through Video Understanding and Reinforcement Learning, and Its Applications to Real Environments) and partly supported by Institute of Information & communications Technology Planning & Evaluation (IITP) grant funded by the Korea government(MSIT) (No.RS-2022-II220184, Development and Study of AI Technologies to Inexpensively Conform to Evolving Policy on Ethics)

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

# Contents

# A Proof of Theorem 1

**Theorem 1** (GradSPO Loss as Upper Bound)**.** *Let $L_{GradSPO}(s_\theta, s_{ref}; T^w, T^l)$ denote the GradSPO loss functional as defined in Eq. 10, where $T^w$ and $T^l$ are the target winning and losing scores. Let $s^w_{ideal} = s_\theta + \gamma \nabla_{\mathbf{x}_t} r(\mathbf{x}_t, t)$ and $s^l_{ideal} = s_\theta - \gamma \nabla_{\mathbf{x}_t} r(\mathbf{x}_t, t)$ be the ideal target scores based on true reward gradients. Let $\hat{s}^w_\theta$ and $\hat{s}^l_\theta$ (as defined in Eq. 9) be the target scores constructed using the approximated, potentially noisy, gradient signal. Then, the GradSPO loss computed with the approximated targets forms an upper bound on the loss computed with ideal targets:*

$$L_{GradSPO}(s_\theta, s_{ref}; s^w_{ideal}, s^l_{ideal}) \leq L_{GradSPO}(s_\theta, s_{ref}; \hat{s}^w_\theta, \hat{s}^l_\theta). \tag{17}$$

*Proof.* From Eq. 8, we have:
$$\mathbb{E}(\gamma'_t z^\pm) = \pm \gamma \, \nabla_{x_t} r(\mathbf{x}_t, t),$$
$$Var(\gamma'_t z^\pm) = I. \tag{18}$$

It follows that:
$$\mathbb{E}(\hat{s}^{w,l}_\theta) = s^{w,l}_{ideal},$$
$$Var(\hat{s}^{w,l}_\theta) = I. \tag{19}$$

Next, consider the GradSPO loss in Eq. 10 (where we omit the stop gradient operator $sg(\cdot)$ for simplicity):

$$
\begin{aligned}
L_{\text{GradSPO}}(s_\theta, s_{\text{ref}}; s^w_{\text{ideal}}, s^l_{\text{ideal}}) = -\mathbb{E}\Big[ \log \sigma\Big( -\beta w(t)\Big( & \|s_\theta(\mathbf{x}_t, \mathbf{c}, t) - s^w_{\text{ideal}}(\mathbf{x}_t, \mathbf{c}, t)\|_2^2 \\
& - \|s_{\text{ref}}(\mathbf{x}_t, \mathbf{c}, t) - s^w_{\text{ideal}}(\mathbf{x}_t, \mathbf{c}, t)\|_2^2 \\
& - \|s_\theta(\mathbf{x}_t, \mathbf{c}, t) - s^l_{\text{ideal}}(\mathbf{x}_t, \mathbf{c}, t)\|_2^2 \\
& + \|s_{\text{ref}}(\mathbf{x}_t, \mathbf{c}, t) - s^l_{\text{ideal}}(\mathbf{x}_t, \mathbf{c}, t)\|_2^2 \Big) \Big) \Big].
\end{aligned}
\tag{20}
$$

Using $\mathbb{E}(\hat{s}^w_\theta) = s^w_{\text{ideal}}$ and $\mathbb{E}(\hat{s}^l_\theta) = s^l_{\text{ideal}}$, we can rewrite the loss as:

$$
\begin{aligned}
L_{\text{GradSPO}}(s_\theta, s_{\text{ref}}; s^w_{\text{ideal}}, s^l_{\text{ideal}}) = -\mathbb{E}\Big[ \log \sigma\Big( -\beta w(t)\Big( & \|s_\theta(\mathbf{x}_t, \mathbf{c}, t) - \mathbb{E}[\hat{s}^w_\theta(\mathbf{x}_t, \mathbf{c}, t)]\|_2^2 \\
& - \|s_{\text{ref}}(\mathbf{x}_t, \mathbf{c}, t) - \mathbb{E}[\hat{s}^w_\theta(\mathbf{x}_t, \mathbf{c}, t)]\|_2^2 \\
& - \|s_\theta(\mathbf{x}_t, \mathbf{c}, t) - \mathbb{E}[\hat{s}^l_\theta(\mathbf{x}_t, \mathbf{c}, t)]\|_2^2 \\
& + \|s_{\text{ref}}(\mathbf{x}_t, \mathbf{c}, t) - \mathbb{E}[\hat{s}^l_\theta(\mathbf{x}_t, \mathbf{c}, t)]\|_2^2 \Big) \Big) \Big],
\end{aligned}
\tag{21}
$$

since

$$\mathbb{E}\big[\|s_{\text{ref}}(\mathbf{x}_t, \mathbf{c}, t) - \hat{s}^w_\theta(\mathbf{x}_t, \mathbf{c}, t)\|_2^2\big] = \|\mathbb{E}[s_{\text{ref}}(\mathbf{x}_t, \mathbf{c}, t) - \hat{s}^w_\theta(\mathbf{x}_t, \mathbf{c}, t)]\|_2^2 + n \, Var(\hat{s}^w_\theta(\mathbf{x}_t, \mathbf{c}, t)),$$

where $n$ denotes the dimension of the variable $\hat{s}_\theta$. Since the variance terms are equal, they cancel each other out, allowing us to move the expectation out of the norm:

$$
\begin{aligned}
L_{\text{GradSPO}}(s_\theta, s_{\text{ref}}; s^w_{\text{ideal}}, s^l_{\text{ideal}}) \quad = -\mathbb{E}\Big[ \log \sigma\Big( -\beta w(t)\big( & \mathbb{E}[\|s_\theta(\mathbf{x}_t, \mathbf{c}, t) - \hat{s}^w_\theta(\mathbf{x}_t, \mathbf{c}, t)\|_2^2] \\
& - \mathbb{E}[\|s_{\text{ref}}(\mathbf{x}_t, \mathbf{c}, t) - \hat{s}^w_\theta(\mathbf{x}_t, \mathbf{c}, t)\|_2^2] \\
& - \mathbb{E}[\|s_\theta(\mathbf{x}_t, \mathbf{c}, t) - \hat{s}^l_\theta(\mathbf{x}_t, \mathbf{c}, t)\|_2^2] \\
& + \mathbb{E}[\|s_{\text{ref}}(\mathbf{x}_t, \mathbf{c}, t) - \hat{s}^l_\theta(\mathbf{x}_t, \mathbf{c}, t)\|_2^2]\big) \Big) \Big].
\end{aligned}
\tag{22}
$$

Finally, applying Jensen's inequality to the convex function $-\log \sigma(\cdot)$ yields

$$
\begin{aligned}
L_{\text{GradSPO}}(s_\theta, s_{\text{ref}}; s_{\text{ideal}}^w, s_{\text{ideal}}^l) \leq -\mathbb{E}\Big[\log \sigma\Big(&-\beta w(t)\big(\|s_\theta(\mathbf{x}_t, \mathbf{c}, t) - \hat{s}_\theta^w(\mathbf{x}_t, \mathbf{c}, t)\|_2^2 \\
&- \|s_{\text{ref}}(\mathbf{x}_t, \mathbf{c}, t) - \hat{s}_\theta^w(\mathbf{x}_t, \mathbf{c}, t)\|_2^2 \\
&- \|s_\theta(\mathbf{x}_t, \mathbf{c}, t) - \hat{s}_\theta^l(\mathbf{x}_t, \mathbf{c}, t)\|_2^2 \\
&+ \|s_{\text{ref}}(\mathbf{x}_t, \mathbf{c}, t) - \hat{s}_\theta^l(\mathbf{x}_t, \mathbf{c}, t)\|_2^2\big)\Big)\Big]
\end{aligned}
\tag{23}
$$

$$
= L_{\text{GradSPO}}(s_\theta, s_{\text{ref}}; \hat{s}_\theta^w, \hat{s}_\theta^l).
$$

Thus, the approximate GradSPO loss serves as an upper bound on the exact variant. $\qquad\square$

## B    Proof of Theorem 2

**Theorem 2** (Variance Reduction of Estimated Gradient Signal). *Let $z^+ = G + z_1$ and $z^- = -G + z_2$, where $G = \sqrt{T}\beta_t \nabla_{\mathbf{x}_t} r(\mathbf{x}_t, t)$ is the true scaled gradient signal, and $z_1, z_2 \sim \mathcal{N}(0, \mathbf{I})$ are independent noise terms. The noise-reduced estimate $z_{\text{max-min}}^\pm$ has variance half that of the original estimation $z^\pm$:*

$$
\text{Var}(z_{\text{max-min}}^\pm) = \frac{\text{Var}(z^\pm)}{2}
\tag{24}
$$

*Proof.* We are given two noisy estimates

$$
z^+ = G + z_1 \quad \text{and} \quad z^- = -G + z_2,
$$

where $z_1, z_2 \sim \mathcal{N}(0, \mathbf{I})$ are independent. Consequently,

$$
z^+ \sim \mathcal{N}(G, \mathbf{I}) \quad \text{and} \quad z^- \sim \mathcal{N}(-G, \mathbf{I}),
$$

so $\text{Var}(z^+) = \text{Var}(z^-) = \mathbf{I}$. We define the noise-reduced estimate

$$
z_{\text{max-min}}^\pm = \pm \frac{z^+ - z^-}{2}.
$$

First, to see that this estimator is unbiased, note

$$
\mathbb{E}[z_{\text{max-min}}^\pm] = \mathbb{E}\Big[\frac{z^+ - z^-}{2}\Big] = \pm\frac{1}{2}\big(\mathbb{E}[z^+] - \mathbb{E}[z^-]\big) = \pm\frac{1}{2}(G - (-G)) = \pm G.
$$

Next, to compute its variance, observe that $z^+$ and $z^-$ are independent, so

$$
\text{Var}(z_{\text{max-min}}^\pm) = \text{Var}\Big(\pm\frac{z^+ - z^-}{2}\Big) = \frac{1}{4}\big(\text{Var}(z^+) + \text{Var}(z^-)\big) = \frac{1}{4}(\mathbf{I} + \mathbf{I}) = \frac{1}{2}\mathbf{I}.
$$

Thus,

$$
\text{Var}(z_{\text{max-min}}^\pm) = \frac{1}{2}\text{Var}(z^+) = \frac{1}{2}\text{Var}(z^-),
$$

showing that the variance of $z_{\text{max-min}}^\pm$ is exactly half the variance of the original noisy estimates $z^+$ or $z^-$. Therefore, $\pm\frac{z^+ - z^-}{2}$ is an unbiased estimator of $\pm G$ whose variance is reduced by a factor of $\frac{1}{2}$ compared to the individual noisy vectors. $\qquad\square$

## C    Link Between GradSPO and SPO

The Step-by-step Preference Optimization (SPO) framework utilizes the Denoising Diffusion Implicit Models (DDIM) [25] transition probability. This probability, $p_\theta(\mathbf{x}_{t-1}|\mathbf{x}_t)$, from $\mathbf{x}_t$ to $\mathbf{x}_{t-1}$ is defined as a Gaussian distribution:

$$
p_\theta(\mathbf{x}_{t-1}|\mathbf{x}_t) = \mathcal{N}\big(\mathbf{x}_{t-1}; \boldsymbol{\mu}_\theta(\mathbf{x}_t, \mathbf{c}, t), \sigma_t^2\mathbf{I}\big),
\tag{25}
$$

where

$$\boldsymbol{\mu}_\theta(\mathbf{x}_t, \mathbf{c}, t) = \sqrt{\bar{\alpha}_{t-1}} \left( \frac{\mathbf{x}_t - \sqrt{1 - \bar{\alpha}_t} \boldsymbol{\epsilon}_\theta(\mathbf{x}_t, \mathbf{c}, t)}{\sqrt{\bar{\alpha}_t}} \right) + \sqrt{1 - \bar{\alpha}_{t-1} - \sigma_t^2} \cdot \boldsymbol{\epsilon}_\theta(\mathbf{x}_t, \mathbf{c}, t). \tag{26}$$

In this formulation, the terms $\sigma_t^2$, $\bar{\alpha}_t$, and $\epsilon_\theta$ follow the definitions provided in Section 2.1. Given these definitions, the log-probability of a specific sample $\mathbf{x}_{t-1}^{w,l}$ can be expressed as:

$$\log p_\theta(\mathbf{x}_{t-1}^{w,l} | \mathbf{x}_t) = -\frac{1}{2\sigma_t^2} \left\| \boldsymbol{\mu}_\theta(\mathbf{x}_t, \mathbf{c}, t) - \mathbf{x}_{t-1}^{w,l} \right\|_2^2 + C, \tag{27}$$

where the constant $C$ is given by $C = -\frac{d}{2} \log \left( 2\pi\sigma_t^2 \right)$, with $d$ being the dimensionality of the variable $\mathbf{x}_t$.

Rewriting the log-probability where $\mathbf{x}_{t-1}^{w,l}$ is expressed using an auxiliary variable $\mathbf{z}^\pm$:

$$\log p_\theta(\mathbf{x}_{t-1}^{w,l} | \mathbf{x}_t) = -\frac{1}{2\sigma_t^2} \left\| \boldsymbol{\mu}_\theta(\mathbf{x}_t, \mathbf{c}, t) - \boldsymbol{\mu}_\theta(\mathbf{x}_t, \mathbf{c}, t) + \sigma_t \mathbf{z}^\pm \right\|_2^2 + C. \tag{28}$$

This expression can be further reformulated in terms of the noise prediction $\epsilon_\theta(\mathbf{x}_t, \mathbf{c}, t)$:

$$\log p_\theta(\mathbf{x}_{t-1}^{w,l} | \mathbf{x}_t) = -A(t) \left\| \boldsymbol{\epsilon}_\theta(\mathbf{x}_t, \mathbf{c}, t) - \left( \boldsymbol{\epsilon}_\theta(\mathbf{x}_t, \mathbf{c}, t) - B(t)\sqrt{1 - \bar{\alpha}_t} \mathbf{z}^\pm \right) \right\|_2^2 + C. \tag{29}$$

The time-dependent coefficients $A(t)$ and $B(t)$ are defined as:

$$A(t) := \frac{1}{2\sigma_t^2} \left( \frac{\sqrt{\bar{\alpha}_{t-1}(1 - \bar{\alpha}_t)}}{\sqrt{\bar{\alpha}_t}} - \sqrt{1 - \bar{\alpha}_{t-1} - \sigma_t^2} \right)^2 \tag{30}$$

$$B(t) := \frac{\sigma_t}{\sqrt{1 - \bar{\alpha}_t} \left( \frac{\sqrt{\bar{\alpha}_{t-1}(1 - \bar{\alpha}_t)}}{\sqrt{\bar{\alpha}_t}} - \sqrt{1 - \bar{\alpha}_{t-1} - \sigma_t^2} \right)}. \tag{31}$$

By comparing the rewritten form of $\log p_\theta(\mathbf{x}_{t-1}^{w,l} | \mathbf{x}_t)$ with the objective function of GradSPO in Eq. 12, it becomes evident that SPO can be viewed as a specific instance of GradSPO. The correspondence is established by setting the GradSPO parameters $a(t)$ and $\gamma_t$ as follows:

$$a(t) = A(t) \text{ and } \gamma_t = B(t). \tag{32}$$

This demonstrates that GradSPO can be considered a generalization of the original SPO method, where SPO emerges under a particular choice for the functions $a(t)$ and $\gamma_t$.

## D  Additional Experimental Details

| Hyperparameters | SD 1.5 | SDXL |
|---|---|---|
| Learning rate | 6e-5 | 1e-5 |
| # of epochs | 10 | 10 |
| Batch size | 40 | 16 |
| $\mu$ | 0.9 | 0.9 |
| $\beta$ | 10 | 10 |
| $\kappa$ | [0, 750] | [0, 750] |
| LoRA rank | 4 | 64 |
| cfg during training | 5.0 | 5.0 |
| # of samples per step | 4 | 4 |
| Sampling steps during training | 20 | 20 |
| GPU Setup | 4x NVIDIA A100 | 4x NVIDIA A100 |

Table 3: Hyperparameter settings used for SD 1.5 and SDXL backbones.

We generally adopted hyperparameters aligned with those used in SPO, training all models with AdamW [38] using a weight decay of 1e-4. Additional hyperparameter details are provided in Table 3. For SDXL, however, we deviated from the standard multi-step preference optimization procedure of SPO, instead employing a single-step approach analogous to the one utilized in SD1.5. Although the multi-step preference optimization typically yields superior outcomes compared to the single-step method within SPO, our single-step procedure for SDXL nonetheless achieved performance surpassing that of SPO.

# E  Limitations

While GradSPO significantly improves upon the original SPO method, our current implementation remains restricted to single-step preference optimization and has not yet been extended to multi-step preference optimization. Nonetheless, GradSPO consistently outperforms SPO, even though SPO employs multi-step preference optimization with models such as SDXL, underscoring the robustness and effectiveness of GradSPO's gradient-based viewpoint. Additionally, although GradSPO training does not necessitate an explicit image dataset, it relies on a prompt dataset and a trained reward model, rendering it susceptible to biases inherent in the training data. These biases can propagate throughout the optimization process, potentially affecting the quality, fairness, and generalizability of outcomes. Despite these limitations, GradSPO provides valuable theoretical insights by generalizing and improving upon SPO, clarifying both the fundamental advantages and inherent challenges associated with stepwise preference-based optimization techniques.

# F  Evaluations on Parti Prompts and HPDv2

This section presents additional comparative evaluations conducted using the Parti Prompts [39] consisting of over 1600 prompts and HPDv2 [21] datasets consisting of 400 prompts. The results,

| Model | Method | HPSv2 | Pickscore | Aesthetic Score | Image Reward |
|---|---|---|---|---|---|
| SD 1.5 [18] | Baseline | 26.29 | 20.63 | 5.2191 | -0.2901 |
| | DPO [6] | 26.71 | 21.22 | 5.4662 | 0.1179 |
| | InPO [8] | 27.60 | 21.62 | 5.6009 | **0.6178** |
| | SPO [10] | 27.25 | 21.65 | 5.8101 | 0.2975 |
| | **GradSPO (Ours)** | **27.70** | **21.93** | **5.8273** | 0.5325 |
| SDXL [19] | Baseline | 27.63 | 22.58 | 5.9639 | 0.6726 |
| | DPO [6] | 28.28 | 23.03 | 6.0127 | 0.9613 |
| | MaPO [13] | 27.92 | 22.66 | 6.0699 | 0.7413 |
| | InPO [8] | 28.78 | 23.18 | 6.0504 | 0.9599 |
| | SPO [10] | 29.00 | 23.75 | 6.1341 | 0.9441 |
| | **GradSPO (Ours)** | **29.55** | **24.26** | **6.3470** | **1.1498** |

**Table 4: HPDv2 results.** Results for GradSPO and preference alignment baselines on HPDv2. For each metric, the top-performing method is **bolded**, while the second-best is underlined.

summarized in Table 4, clearly illustrate the superior performance of GradSPO on the HPDv2 dataset across both SD1.5 and SDXL models. GradSPO significantly outperforms the baseline SPO method, achieving notably higher average scores—29.55 for HPSv2 and 24.26 for PickScore—which underscores its efficacy in enhancing alignment with human preferences.

| Model | Method | HPSv2 | Pickscore | Aesthetic Score | Image Reward |
|---|---|---|---|---|---|
| SD 1.5 [18] | Baseline | 26.50 | 21.27 | 5.1481 | 0.0505 |
| | DPO [6] | 26.78 | 21.56 | 5.2137 | 0.2274 |
| | InPO [8] | **27.62** | 21.85 | 5.4610 | **0.6067** |
| | SPO [10] | 27.16 | 21.72 | 5.5289 | 0.3986 |
| | **GradSPO (Ours)** | 27.49 | **21.99** | **5.6794** | 0.5951 |
| SDXL [19] | Baseline | 27.38 | 22.24 | 5.5929 | 0.4864 |
| | DPO [6] | 28.29 | 22.83 | 5.6826 | 0.9943 |
| | MaPO [13] | 27.72 | 22.41 | 5.7819 | 0.6793 |
| | InPO [8] | 28.48 | 22.87 | 5.7093 | 0.9252 |
| | SPO [10] | 28.78 | 23.32 | 6.0509 | 0.9762 |
| | **GradSPO (Ours)** | **29.35** | **23.80** | **6.1202** | **1.1005** |

**Table 5: Parti prompt results.** Comparison of GradSPO and preference alignment baselines evaluated on Parti prompts. The best-performing method for each metric is highlighted in **bold**, and the runner-up is underlined.

As detailed in Table 5, GradSPO performs comparably to the leading method, InPO, when evaluated with the SD1.5 model on the Parti Prompts dataset, showing only minor differences. Crucially, GradSPO consistently outperforms SPO, highlighting the advantages of our proposed gradient-based optimization approach. Furthermore, when using the SDXL model with the Parti Prompts dataset, GradSPO surpasses all baseline methods, confirming its robust ability to generate visually appealing images.

Overall, GradSPO achieves state-of-the-art or highly competitive results across various reward metrics for both datasets. The consistent and significant improvements over the original SPO variant further validate the effectiveness and practical benefits of viewing SPO from this gradient-based optimization perspective.

## G Comparisons with Clean Gradient

In this section, we present an ablation study comparing GradSPO to a variant trained with exact gradients, which we term "Clean Gradient." Unlike GradSPO, which uses a noisy gradient approximation, Clean Gradient utilizes exact gradients obtained by backpropagating through the reward model. We observed instability when training with gradients with respect to $\mathbf{x}_t$ (i.e., $\nabla_{\mathbf{x}_t}$), consistent with prior findings [40, 41]. Consequently, for our Clean Gradient experiments, we used gradients with respect to the estimated clean latent, $\mathbf{x}_0$ (i.e., $\nabla_{\mathbf{x}_0}$), for the Clean Gradient experiments.

| Method | Comp. Cost | | Reward Metrics | | | |
|---|---|---|---|---|---|---|
| | GPU Mem. (GB) | Time (s) | HPSv2 | Pickscore | Aesthetic Score | Image Reward |
| Clean Gradient | 52 | 25 | **27.11** | **21.67** | 5.7058 | **0.5211** |
| GradSPO | **28** | **17** | 26.86 | 21.38 | **5.7651** | 0.4747 |

**Table 6:** Comparison of computational costs and performance metrics for GradSPO and Clean Gradient on Pick-a-Pic v2. Best results per metric are in **bold**. Computational costs per optimization step were measured using a single NVIDIA A100 GPU with a batch size of 5.

Table 6 demonstrates that Clean Gradients generally outperforms GradSPO across most metrics, with the notable exception of the aesthetic score. We primarily attribute this performance gap to the inherent noise in GradSPO's gradient estimation. This highlights the critical role of noise reduction strategies of GradSPO in narrowing this performance difference. *Despite Clean Gradient's better performance, Clean Gradient incurs a higher memory cost, as it requires backpropagation through both the VAE and the reward model (see Table 6).* In contrast, GradSPO offers a more memory-efficient alternative that remains applicable even when the reward model isn't explicitly differentiable.

## H Connection Between Maximal/Minimal Noise and Reward Gradient

This section briefly explains the connection between the maximal and minimal noise, $z^{\pm}$, and the gradient of a reward model, $\nabla_{x_t} r(\mathbf{x}_t, t)$. From an energy-based perspective, the reward function models the conditional likelihood:

$$p(y \mid \mathbf{x}_t) = \frac{e^{\lambda r(\mathbf{x}_t, t)}}{Z}, \tag{33}$$

where $Z$ is the normalization factor and $\lambda$ serves as a guidance scale.

From an optimal control viewpoint, guiding the model to sample from $p(y \mid \mathbf{x}_t)$ equates to sampling from the augmented stochastic differential equation (SDE):

$$d\mathbf{x}_t = \left[-\frac{1}{2}T\beta_t\mathbf{x}_t - T\beta_t\nabla_{\mathbf{x}_t}\log p(\mathbf{x}_t)\right]dt + \sqrt{T\beta_t}\left(u^*(\mathbf{x}_t, t)dt + dw\right), \tag{34}$$

where $u^*(\mathbf{x}_t, t)$ denotes the optimal control function minimizing a cost function (negative of the terminal reward), and $dw$ denotes standard Brownian motion.

As detailed in Huang et al. [14] (Section 4.4), this optimal control function is connected to the reward gradient:

$$u^*(\mathbf{x}_t, t) = \sqrt{T\beta_t} \nabla_{\mathbf{x}_t} \log p(y \mid \mathbf{x}_t)$$
$$= \lambda \sqrt{T\beta_t} \nabla_{\mathbf{x}_t} r(\mathbf{x}_t, t). \tag{35}$$

Additionally, the noise maximizing this reward (Eq. 7), z + , provides an upper bound to this optimal control function, $u^*(\mathbf{x}_t, t)$. Hence, [14] employs it as an approximation for the sum of control and Brownian increments (see Section 4.3 of [14]):

$$(u^*(\mathbf{x}_t, t) + z)dt \approx z^+ dt. \tag{36}$$

A similar reasoning can be applied to noise minimizing the reward, approximating the negative reward gradient $\nabla_{\mathbf{x}_t} r(\mathbf{x}_t, t)$ with minimal noise $z^-$.

## I GenEval Results

To further assess the compositional and perceptual capabilities of our models, we evaluate them on the GenEval benchmark using SDXL-based backbones. As shown in Table 7, GradSPO achieves notable gains over SPO—not only in aesthetic quality but also in compositional fidelity—resulting in a higher overall score. However, consistent with prior findings from the SPO study, GradSPO still falls short of Diff-DPO in overall GenEval performance. This suggests that, like SPO, GradSPO tends to prioritize aesthetic reward signals over fine-grained compositional alignment, despite outperforming Diff-DPO on all individual image-level reward metrics.

| Method | Single | Two | Count | Color | Pos | AttrBind | Overall |
|---|---|---|---|---|---|---|---|
| SDXL | 97.81 | 68.43 | 40.62 | 86.70 | 12.00 | 23.00 | 54.76 |
| Diff-DPO | **99.69** | **81.06** | **48.44** | **89.63** | **13.25** | **27.75** | **59.84** |
| SPO | 97.81 | 73.74 | 41.25 | 86.44 | 13.00 | 20.25 | 55.41 |
| GradSPO | 99.06 | 77.78 | 47.50 | 88.03 | 13.00 | 22.25 | 57.07 |

**Table 7:** GenEval results for SDXL models. Each column reports accuracy (%) for a specific evaluation dimension, and the final column shows the overall GenEval score.

## J Effect of Timestep Weighting

To investigate the impact of different timestep weighting strategies on preference-based diffusion training, we compare three commonly used schemes: *min-SNR* weighting [42], *P2* weighting [43], and uniform weighting. Table 8 presents results for GradSPO trained on SD 1.5 using the Pick-a-Pic v2 dataset.

Unlike standard diffusion training—where non-uniform schemes such as min-SNR or P2 often outperform uniform weighting—we observe that uniform weighting performs slightly better across most preference-learning metrics. This indicates that, although these weighting schemes have been widely adopted for diffusion model training, their effectiveness does not directly transfer to the preference optimization setting.

We hypothesize that this discrepancy arises from the differing objectives of diffusion generation and diffusion preference learning. While the former seeks to approximate a data distribution, preference learning focuses on maximizing the margin between preferred and non-preferred samples. Consequently, uniform weighting may provide a better inductive bias for stable optimization in margin-based objectives.

## K Additional Qualitative Results

Figure 6 and Figure 7 present additional image samples generated by GradSPO and various preference alignment methods on SDXL for qualitative comparisons on Parti Prompts and HPDv2, respectively.

| Weighting | HSPv2 | Pick Score | Aesthetic Score | Image Reward |
|---|---|---|---|---|
| Min-SNR [42] | 26.72 | 21.26 | 5.6590 | 0.5165 |
| P2 [43] | 26.79 | 21.25 | 5.6028 | **0.5318** |
| Uniform | **26.86** | **21.38** | **5.7651** | 0.4747 |

**Table 8:** Comparison of timestep weighting schemes for GradSPO trained on SD 1.5 with the Pick-a-Pic v2 dataset. The best result in each column is highlighted in bold.

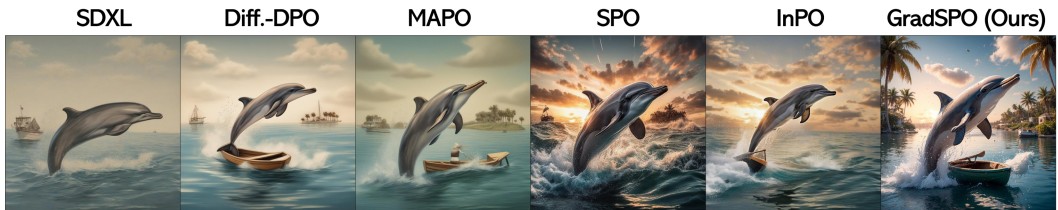

a dolphin jumping over a rowboat

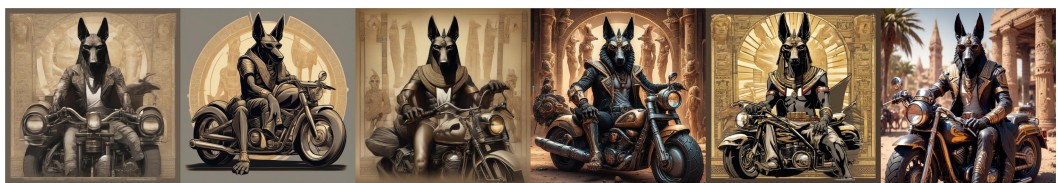

Anubis wearing sunglasses and sitting astride a hog motorcyle

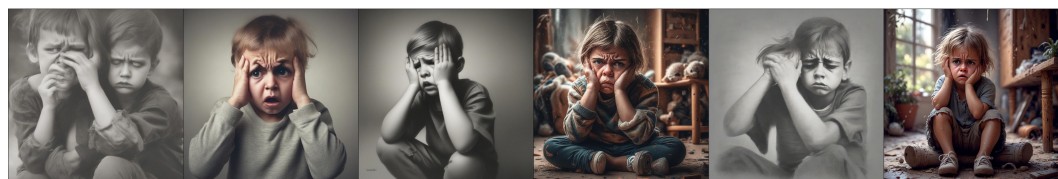

a frustrated child

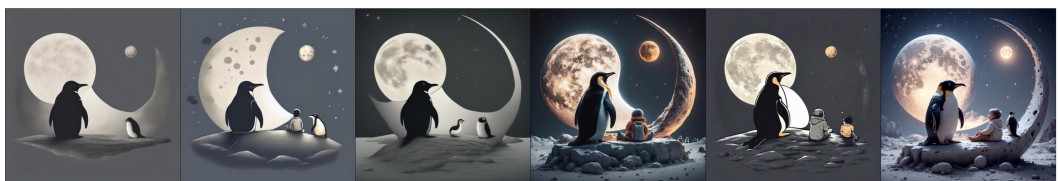

a child and a penguin sitting on the moon

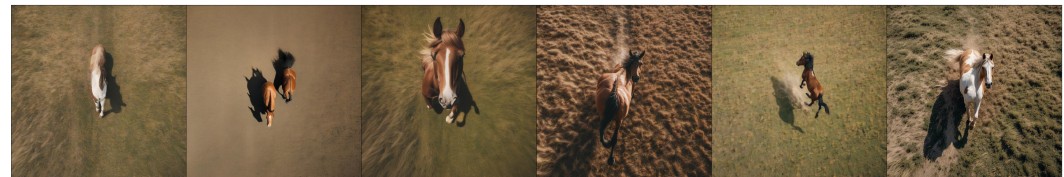

a top down view of a horse running in a field

**Figure 6: Side-by-side comparison of images generated by related methods on Parti Prompts using SDXL.**

SDXL          Diff.-DPO          MAPO          SPO          InPO          GradSPO (Ours)

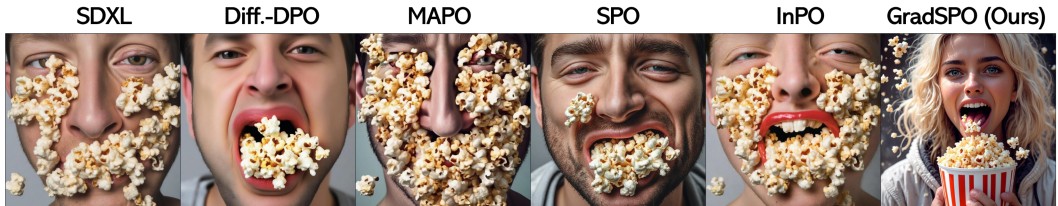

Popcorn in mouth.

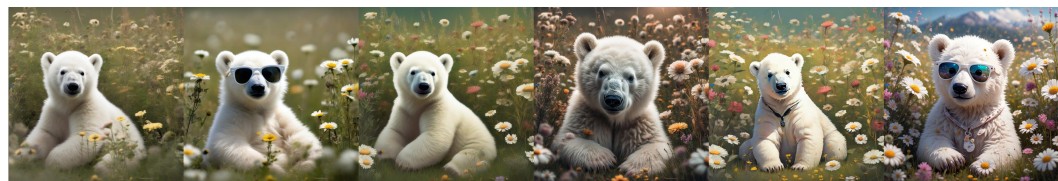

A white polar bear cub wearing sunglasses sits in a meadow with flowers.

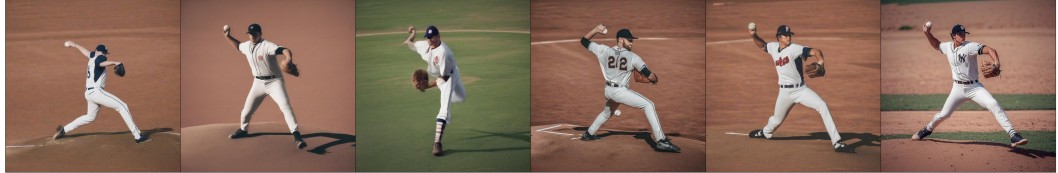

A baseball player pitching a baseball on a field.

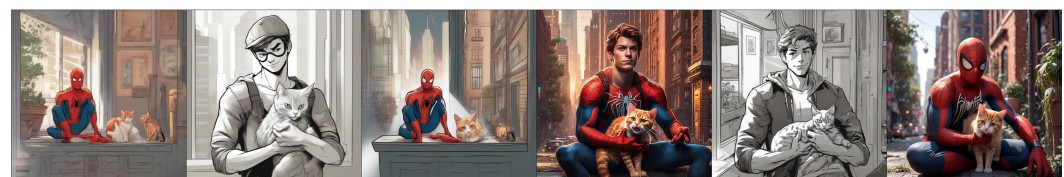

Spider-Man holding a ginger cat

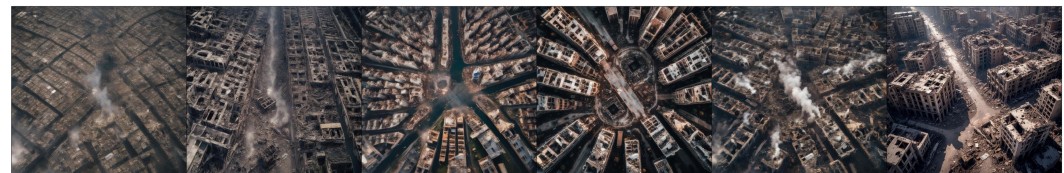

Looking down at a destroyed city from a plane

**Figure 7: Side-by-side comparison of images generated by related methods on HPDv2 using SDXL.**

