# OpenReview forum: "A Gradient Guidance Perspective on Stepwise Preference Optimization for Diffusion Models"
_NeurIPS.cc/2025/Conference — NeurIPS 2025 poster_

### Official Review · Reviewer_VBZ5 · 2025-06-30

**Clarity:** 3
**Significance:** 2
**Originality:** 3
**Rating:** 3
**Confidence:** 4

**Summary:**

This paper introduces GradSPO, a novel framework that reinterprets Stepwise Preference Optimization (SPO) for aligning text-to-image diffusion models with human preferences through a gradient guidance perspective. While SPO improves upon Direct Preference Optimization (DPO) by leveraging intermediate diffusion steps for preference learning, the authors identify two key limitations: (1) an imbalanced temporal weighting that under-prioritizes later denoising steps critical for fine details, and (2) high noise in gradient estimates from reward models. GradSPO addresses these issues by reformulating SPO’s mechanism as gradient-based score matching, leading to a simplified loss with uniform timestep weighting and a variance-aware noise reduction strategy to stabilize training. Experiments on Stable Diffusion 1.5 and SDXL demonstrate that GradSPO outperforms existing methods (e.g., DPO, SPO) in human preference alignment, generating images with enhanced aesthetic quality and prompt faithfulness.

**Questions:**

No

**Ethical Concerns:**

["NO or VERY MINOR ethics concerns only"]

**Final Justification:**

Thank you for your detailed response. The clarification is neat and clear, but after considering the many missing experiments and ablations in the paper, I would recommend authors to thoroughly revise thier paper for a next submission. I would maintain my score.

**Limitations:**

See Weaknesses

**Quality:**

2

**Strengths And Weaknesses:**

Strengths:

1. Theoretical Rigor: The paper provides a solid theoretical foundation for GradSPO, rigorously addressing the underweighting issue in SPO. The derivations are clear and well-justified, making a convincing case for the proposed method from a theoretical perspective.
2. Problem Identification: The authors correctly identify a potential limitation in SPO—underweighting later generative steps—and propose a principled solution. This is a meaningful contribution, as later diffusion steps are crucial for high-frequency detail refinement.
3. Novelty in Formulation: GradSPO introduces a new gradient-based weighting scheme, which is a non-trivial extension of SPO. The idea of dynamically adjusting step weights to stabilize training is theoretically appealing.

Weaknesses:

1. Lack of Empirical Validation:
   1. The paper claims that underweighting later steps harms output quality (e.g., blurriness, incoherence), but no fine-grained analysis (e.g., per-step contribution to image sharpness, edge quality) is provided.
   2. The claim that GradSPO improves training stability is unsupported by experiments. There are no training curves, loss variance comparisons, or reward convergence plots between SPO and GradSPO.
2. Weak Experimental Section:
   1. The experiments focus primarily on downstream metrics (e.g., FID, IS) without isolating the impact of GradSPO’s weighting mechanism. It’s unclear whether improvements stem from better weighting or other factors.
   2. No ablation studies (e.g., removing GradSPO’s gradient correction) to validate its necessity.
3. Overstated Practical Impact:
   1. The theoretical benefits of GradSPO are not matched by empirical gains. The performance improvements over SPO (if any) are marginal in the presented results.
   2. The paper does not explore failure cases or scenarios where GradSPO might degrade performance (e.g., noisy gradients exacerbating instability).

---

> ### Author Rebuttal · Authors · 2025-07-31
>
> Thank you for taking the time to review our submission and provide detailed feedback. We’ve addressed your comments below in the order they were raised.
>
> ---
> ### **Weaknesses (W)**
>
> **[W-Q1.1] Lack of empirical justification for timestep bias problem raised**
>
> **[W-A1.1]** To clarify the effect of timestep bias, we conducted an additional empirical study. Specifically, we plotted the change in preference scores of the $𝑥_0$ predictions at each timestep before and after preference fine-tuning, for both SPO and GradSPO, using noisy images from the Win subset of the Pick-a-Pic v2 dataset on Stable Diffusion 1.5.
>
> The results show that SPO significantly improves preference scores in early timesteps, but the improvements sharply decline in later steps, indicating an exponential bias toward early timesteps. In contrast, GradSPO exhibits a more gradual, linear decline, suggesting it distributes learning improvements more evenly across timesteps.
>
> This supports the presence of timestep bias in SPO and highlights GradSPO's advantage. ***The corresponding plot and discussions will be included in the main paper.***
>
> **[W-Q1.2] Lack of empirical results supporting improved training stability claims of GradSPO**
>
> **[W-A1.2]** We thank the reviewer for highlighting this point. To support our claim regarding GradSPO’s improved training stability, we will include a training reward convergence curve in the camera-ready version. This curve demonstrates that GradSPO consistently achieves higher training rewards compared to SPO, which we attribute to reduced gradient noise and more stable optimization. We will also revise the introduction and abstract to clarify our definition of '*training stability*' and better align the narrative with the empirical evidence presented.
>
> **[W-Q2.1] Lack of ablation experiments isolating the weighting mechanism**
>
> **[W-A2.1]** In Section 4.4 (Figure 5) for our main paper, we provide an ablation study to isolate the weighting mechanism of GradSPO. Specifically, we introduce a simplified objective that incorporates only the new weighting mechanism to standard SPO (*denoted as SPO+Simple*), to better assess the individual contribution of the weighting mechanism. Our results show that the weighting mechanism contributes positively, leading to improved win rates over the baseline SPO on the SD 1.5 model.
>
> **[W-Q2.2] Lack of ablation experiments for gradient correction**
>
> **[W-A2.2]** An ablation study analyzing the effect of the gradient correction term is also provided in Section 4.4 (Figure 5) (*denoted as SPO+Simple+Maxmin*). It shows a clear performance gain over the simplified objective alone (SPO+Simple), demonstrating the contribution of each component to the overall effectiveness of GradSPO*.
>
> *(note that GradSPO would be equivalent to *SPO+Simple+Maxmin+EMA*).
>
> **[W-Q3.1] The theoretical benefits of GradSPO are not matched by empirical gains.**
>
> **[W-A3.1]** We respectfully disagree with this assessment. GradSPO shows consistent improvements over SPO across all benchmarks. While the absolute gains (~0.2 across preference metrics) may seem modest, they are in fact meaningful, especially when compared to absolute gains of DPO to SPO on both SD 1.5 and SDXL.
>
> These results indicate that GradSPO not only offers theoretical advantages but also translates them into empirical gains, reinforcing its effectiveness as a practical preference learning approach.
>
> **[W-Q3.2] The paper does not explore failure cases**
>
> **[W-A3.2]** To better understand the failure cases in terms of training of GradSPO, we vary the $\gamma$ parameter on SD1.5 on Pick-a-Pic v2 while keeping the rest fixed to the setting used in the main results.
>
> **Table A** summarizes the effect of varying $\gamma$ on key metrics such as HSPv2, Pick Score, Aesthetic Score, and Image Reward. The results suggest that at higher guidance scales, increasing $\gamma$ introduces more variance due to noise, leading to worse convergence. Conversely, lower $\gamma$ values result in a weaker reward signal, preventing the model from optimizing effectively. We find that $\gamma = 0.5$ offers a good trade-off between stability and reward strength, and we adopt this setting in both SDXL and SD1.5 experiments.
>
> **Table A.** Ablation study on reward guidance scale $\gamma$ for GradSPO on SD 1.5 for Pick-a-Pic v2.
> | Gamma | HSPv2 | Pick Score | Aesthetic Score | Image Reward |
> |-------|--------|-------------|------------------|----------------|
> | 0.25  | 26.67 | 20.71      | 5.5278           | 0.452          |
> | 0.5   | 26.86 | 21.38      | 5.7651           | 0.4747         |
> | 0.75  | 26.70  | 20.99      | 5.463            | 0.3878         |
>
>
> Moreoever, one potential failure area worth exploring in future work is the reduction in diversity after alignment. Specifically, we observe that stronger alignment to the preference signal tends to reduce output diversity. To investigate this, we measured diversity scores across 10 randomly sampled prompts with 10 different seeds each, following the setup in [1].
>
> The results show a consistent drop in diversity: SDXL scores *12.54*, SPO drops to *10.32*, and GradSPO further to *9.40*. This mirrors patterns seen in preference learning for large language models [2], where optimization toward high-reward outputs often comes at the cost of diversity. Understanding and mitigating this trade-off remains an open research challenge.
>
> > [1] Ibarrola et al., Measuring Diversity in Co-creative Image Generation
>
> > [2] Cui et al., The Entropy Mechanism of Reinforcement Learning for Reasoning Language Models

---

### Official Review · Reviewer_ccWi · 2025-07-02

**Clarity:** 3
**Significance:** 3
**Originality:** 3
**Rating:** 4
**Confidence:** 4

**Summary:**

The paper proposes GradSPO, a follow-up to the SPO approach for preference optimization and text-to-image alignment. Specifically, instead of optimizing likelihood during denoising steps it directly approximates reward gradients using noise vectors that produce the best and worst samples and constructs guided targets by combining these gradient directions.
The key contributions of the paper are:
- Theoretical contribution showing SPO can be interpreted as gradient based score matching.
- GradSPO adds a timestep-uniform objective that removes the need for heuristic weighting.
- Adds a max-min noise estimator to reduce variance in the gradient leading to more stable optimization.
- EMA-based target smoothing which stabilizes training.
- SOTA results across multiple aesthetic benchmarks, datasets and models.

**Questions:**

- The fact that the author did not have to implement "multi-step" SPO to make SDXL work is very interesting but under-explored in the paper.

- For the method to stand up on its own it is important to consider redoing key ablation studies that could have changed from SPO: such as number of timesteps, guidance scale, and some new parameters (like /mu).

- Adding specific failure cases would be very interesting to examine, especially if they are not shared with SPO, which can inspire future research into the approach.

 - The authors should also consider evaluating on GenEval results which are missing here but are in the SPO paper

- A figure would be very useful to show a comparison of the GradSPO and SPO methods as a quick reference and introduction to the paper.

- I found Figure 4 confusing. The side-by-side comparisons expected typically for the preference models is similar to Figures 6 - 7 in the Appendix, meaning the difference is clearly visible, but the images are relatively the same. Doing some testing on SDXL and DPO demos for the prompts specified: a) "Buff Harry Potter" and b) "a gorgeous queen with cat like eyes", I was able to get a correct figure for a) SDXL (6/10), DPO (10/10) b) SDXL (9/10), DPO (8/10). These are not rigorous tests,but they suggest handpicked results or preferential selection for GradSPO. I do not have access to GradSPO directly so I cannot perform a fair evaluation. However, the demos demonstrated for the GradSPO in Figure 4 are better even among the ones I sampled so I do not see a reason why the authors would pick poor images for the other methods, which damages the credibility of the comparison. I suggest revising the qualitative demos in Figure 4 to match the extended ones in the Appendix. For a fairer comparison multiple demos can be sampled for each model, scored and the best one for each method selected.

- The user evaluation also raises concerns. Human preference studies are easy to bias via position effects or presentation order. The performance of GradSPO vs SDXL (~90% win rate) is a unusually high for aesthetic alignment so I urge the authors to describe their process and guarantee all the necessary steps have been taken for a fair comparison. It would also be better to follow SPO’s breakdown and perform a user study based on General Preference, Visual Appeal and Prompt Alignment categories.

- Minor point: the ablation studies in Section 4.4 make a strong argument in favor of each component in isolation. However, winrate is a non transitive metric so SPO>SDXL and GradSPO>>SDXL does not mean GradSPO>>SPO, so a winrate vs SPO or a table with scores is more informative.

**Ethical Concerns:**

["NO or VERY MINOR ethics concerns only"]

**Final Justification:**

As mentioned in the discussion my main concerns revolved around the comparison with other models which have been addressed and I am thus increasing my score.

**Limitations:**

Addressed

**Quality:**

2

**Strengths And Weaknesses:**

Strengths
- Clear theoretical motivation for interpreting SPO as score matching.
- Simple objective that avoids timestep-specific weighting, noise variance reduction and EMA smoothing.
- GradSPO has good results results across multiple aesthetic benchmarks and ablation studies show each component is well motivated.
- GradSPO is robust and works without requiring multi-step implementation which is central to SPO.

Weaknesses
- The method relies heavily on SPO but does not re-examine key hyperparameters that were ablated in SPO (see Questions).
- No visual side-by-side comparison with SPO which could have been useful to demonstrate the differences.
- SPO evaluations are missing and key issues with qualitative results and user studies (see Questions)
- Lack of specific qualitative failure cases to examine the method in detail and guide future improvements.

---

> ### Author Rebuttal · Authors · 2025-07-31
>
> We thank the reviewer for the insightful comments and careful evaluation. Our responses are organized according to the sequence of points raised in the review.
>
> ### **Weaknesses (W)**
> **[W-Q1] addressed in [Q-Q2] below**
>
> **[W-Q2] No visual side-by-side comparison with SPO.**
>
> **[W-A2]** Side-by-side visual comparisons with SPO are already included in Figure 4 of the main paper, as well as Figures 6–7 in the Appendix. These figures present qualitative comparisons between GradSPO, SPO, and other widely used preference learning methods to highlight their differences more clearly.
>
> **[W-Q3] addressed in [Q-Q6] and [Q-Q7] below**
>
> **[W-Q4] addressed in [Q-Q3] below**
>
>
> ---
> ### **Questions (Q)**
> **[Q-Q1] The fact that the author did not have to implement "multi-step" SPO to make SDXL work is very interesting but under-explored in the paper.**
>
> **[Q-A1]** 'Multi-step' SPO was originally introduced to strengthen reward signals by simulating longer trajectories. However, in GradSPO, the use of noise reduction stabilizes gradient estimates, effectively reducing the need for such rollouts. This allows SDXL to achieve strong performance without the added complexity of multi-step simulation, highlighting noise reduction as a more effective alternative in this setting.
>
> **[Q-Q2] Lack of ablation on key hyperparameters.**
>
> **[Q-A2]** In **Table A**, we present ablation results of our GradSPO on SPO-specific hyperparameters, including guidance scale and timestep count for SD 1.5 on Pick-a-Pic v2 with all other hyperparameters fixed to the ones used in the main results.
>
> We observe that increasing the guidance scale leads to a consistent drop in performance, while reducing it improves results. We hypothesize this is due to high CFG scales introducing oversaturation artifacts during sampling [1]. While high CFG can yield visually appealing images during inference, training with these settings amplifies artifacts, ultimately degrading generation quality.
>
> We also examine the impact of increasing the number of sampling steps from 20 to 50. As shown in the table, this change yields no meaningful performance improvement, with results remaining comparable to those obtained at 20 steps. Additionally, it nearly doubles the training time due to the increased sampling overhead.
>
> **Table A.** Ablation study on classifier-free guidance scale and number of timesteps in GradSPO on SD 1.5.
> |Guidance|HSPv2|Pick Score|Aesthetic Score|Image Reward|
> |-|-|-|-|-|
> |7.5|26.60|21.19|5.5051|0.3066|
> |5.0|26.86|21.38|5.7651|0.4747|
> |4.0|27.19|21.44|5.7669|0.5852|
>
> |Steps|HSPv2|Pick Score|Aesthetic Score|Image Reward|
> |-|-|-|-|-|
> |20|26.86|21.38|5.7651|0.4747|
> |50|26.87|21.21|5.7939|0.4563|
>
> Additionally, in **Table B**, we conduct ablations on GradSPO specific hyperparameters, $\gamma$ and $\mu$, with all other settings fixed for SD 1.5 with results reported on Pick-a-Pic v2 (we set $\gamma=0.5$ when varying $\mu$ and $\mu=0.9$ when varying $\gamma$).
>
> **Table B.** Ablation results on reward guidance scale $\gamma$ and EMA momentum $\mu$ for GradSPO on SD 1.5.
> | $\gamma$ |HSPv2|Pick Score|Aesthetic Score|Image Reward|
> |-|-|-|-|-|
> |0.25|26.67|20.71|5.5278|0.452|
> |0.5|26.86|21.38|5.7651|0.4747|
> |0.75|26.70|20.99|5.463|0.3878|
>
> |$\mu$|HSPv2|Pick Score|Aesthetic Score|Image Reward|
> |-|-|-|-|-|
> |0.9|26.86|21.38|5.7651|0.4747|
> |0.95|26.74|21.12|5.6352|0.4153|
> |0.99|26.68|21.22|5.7591|0.3871|
>
> We find that the $\mu$ parameter is relatively robust, with only a slight drop in performance at lower values. This may be attributed to oversmoothing, where the EMA model updates too slowly. In contrast, modifying the $\gamma$ parameter in either direction results in noticeable performance degradation.
>
> At higher guidance scales, increasing $\gamma$ amplifies noise, leading to higher gradient variance and poorer convergence. Conversely, lowering $\gamma$ weakens the reward signal, preventing the model from optimizing effectively. Overall, we find that $\gamma = 0.5$ provides a strong balance between reward strength and signal stability, and we adopt this value for both SDXL and SD 1.5 experiments.
>
> > [1] Sadat et al., Eliminating Oversaturation and Artifacts of High Guidance Scales in Diffusion Models
>
> **[Q-Q3] Adding specific failure cases would be very interesting.**
>
> **[W-A3]** One potential failure area worth exploring in future work is the reduction in diversity after alignment. Specifically, we observe that stronger alignment to the preference signal tends to reduce output diversity. To investigate this, we measured diversity scores across 10 randomly sampled prompts with 10 different seeds each, following the setup in [1].
>
> The results show a consistent drop in diversity: SDXL scores *12.54*, SPO drops to *10.32*, and GradSPO further to *9.40*. This mirrors patterns seen in preference learning for large language models [2], where optimization toward high-reward outputs often comes at the cost of diversity. Understanding and mitigating this trade-off remains an open research challenge.
>
> > [1] Ibarrola et al., Measuring Diversity in Co-creative Image Generation
>
> > [2] Cui et al., The Entropy Mechanism of Reinforcement Learning for Reasoning Language Models
>
>
> **[Q-Q4] The authors should also consider evaluating on GenEval results.**
>
>
> **[Q-A4]** **Table C.** below presents the GenEval results for SDXL. From these results, we observe that GradSPO achieves notable improvements over SPO, not only in aesthetic quality but also in compositional generation, resulting in a higher overall score. However, similar to what was reported in the SPO paper, GradSPO still falls short of Diff-DPO in overall GenEval performance. This suggests that, similar to SPO, GradSPO tends to prioritize optimizing for aesthetic reward signals over learning fine-grained compositional alignment, despite outperforming Diff-DPO on all individual image reward metrics.
>
> **Table C.** GenEval results for SDXL models.
> |Method|Single|Two|Count|Color|Pos|AttrBind|Overall|
> |-|-|-|-|-|-|-|-|
> |SDXL|97.81|68.43|40.62|86.70|12.00|23.00|54.76|
> |Diff.-DPO|**99.69**|**81.06**|**48.44**|**89.63**|**13.25**|**27.75**|**59.84**|
> |SPO|97.81|73.74|41.25|86.44|13.00|20.25|55.41|
> |GradSPO|99.06|77.78|47.50|88.03|13.00|22.25|57.07|
>
>
> **[Q-Q5] A figure comparing GradSPO and SPO would be helpful as a quick reference in the introduction.**
>
> **[Q-A5]** We recognize that including such a comparison earlier in the paper could improve clarity and accessibility. For the camera-ready version, we will revise the introduction to feature a clearer visual reference as a quick overview of the improvements introduced by GradSPO.
>
>
> **[Q-Q6] Curation details for Figure 4**
>
> **[Q-A6]** We appreciate the reviewer’s careful analysis and thoughtful suggestion. We acknowledge that without a clearly defined sampling or scoring procedure, the presentation may give the impression of preferential selection.
>
> To address this, we will revise Figure 4 in the camera-ready version to follow a more rigorous and transparent protocol. Specifically, we will sample multiple generations per model for each prompt, score them using consistent preference criteria (e.g., reward model scores), and display the top result per method. This should provide a more credible and balanced qualitative comparison. We will also clarify this selection process in the caption and main text.
>
> **[Q-Q7] Concerns regarding user study**
>
> **[Q-A7]** In our original user study, participants were shown image pairs (Image A and Image B) generated using the same prompt and initial noise, each corresponding to a different method. Participants were asked to choose the image they preferred. To mitigate presentation order bias, the assignment of methods to Image A/B was randomized for each comparison. The high win rate in the user study is due to the lack of a "tie" option, which may have inflated the win rate of GradSPO over SDXL by forcing a decision even when the quality was comparable.
>
> To address this, we conducted an additional fine-grained user study with 100 prompt samples per method and 5 independent raters. In this follow-up, participants were allowed to select one of three options: *Image A better*, *Image B better*, or *Tie*. Additionally, similar to SPO, the preference was done over General Preference, Visual Appeal and Prompt Alignment (**Table D** and **Table E**).
>
> **Table D.** User preference results: **GradSPO vs SDXL**.
> |Category|Win|Tie|Lose|
> |-|-|-|-|
> |Overall|75.62%|5.72%|18.66%|
> |Instruction|53.23%|30.85%|15.92%|
> |Aesthetic|81.09%|3.98%|14.93%|
>
> **Table E.** User preference results: **GradSPO vs SPO**.
> |Category|Win|Tie|Lose|
> |-|-|-|-|
> |Overall|52.99%|21.89%|25.12%|
> |Instruction|26.37%|54.98%|18.66%|
> |Aesthetic|56.22%|21.14%|22.64%|
>
> These results confirm that GradSPO maintains a strong preference margin over SDXL and SPO, even with the tie option included, further validating the perceptual improvements introduced by our method.
>
> **[Q-Q8] Non-transitivity of win rates and need for direct comparison with SPO**
>
> **[Q-A8]** To address the concern regarding the non-transitivity of win rates, we provide a direct comparison of each ablation variant against SPO using standard preference metrics in **Table F**. The results show a consistent upward trend as each component is added, reinforcing the value of the proposed design choices in GradSPO.
>
> **Table F.** Preference metrics comparing ablation variants of GradSPO against SPO on SDXL. Gradual inclusion of key components (e.g., max-min guidance) leads to consistent improvements across all metrics.
> |Method|HSPv2|Pick Score|Aesthetic|Image Reward|Avg|
> |-|-|-|-|-|-|
> |SPO + Simple|0.518|0.570|0.646|0.624|0.5895|
> |SPO + Simple + MaxMin| 0.618 | 0.566 |0.718|0.650|0.6380|
> |GradSPO|0.604|0.648|0.800|0.684|0.6840|
>
> ---

---

> > ### Comment · Reviewer_ccWi · 2025-08-05
> >
> > Thank you for the additional clarifications and experiments. My main concern revolved around the experiments and ensuring a fair comparison which the authors have addressed. I am willing to increase my score.

---

### Official Review · Reviewer_XP6v · 2025-07-03

**Clarity:** 3
**Significance:** 2
**Originality:** 2
**Rating:** 4
**Confidence:** 3

**Summary:**

This paper proposes GradSPO, a method designed to alleviate the time-step weighting bias inherent in SPO. The authors support their approach with theoretical analysis and further validate it through experiments on both SD1.5 and SDXL backbones.

**Questions:**

As highlighted in the weaknesses section, the primary concern centers around the simplicity of the proposed function a(t). While the idea is intuitive, the paper does not provide any ablation studies or empirical analysis to justify this specific design choice. Without such evidence, it remains unclear whether the performance gains are inherently tied to this formulation or if alternative strategies—such as more complex or learnable functions—could yield better results.
Furthermore, the absence of robustness checks raises questions about the general applicability and stability of the method under different settings or perturbations.

**Ethical Concerns:**

["NO or VERY MINOR ethics concerns only"]

**Final Justification:**

I appreciate the authors for the helpful feedback. My main concerns are addressed and I would raise my final rating to 4. borderline accept.

**Limitations:**

yes

**Paper Formatting Concerns:**

this paper is well formatted.

**Quality:**

3

**Strengths And Weaknesses:**

Strengths
1. The paper is clearly written and logically organized, making it easy to follow the proposed methodology and its motivations.
2. The GradSPO framework is theoretically well-founded, with analytical derivations that support its design and justify its components.
3. The method’s effectiveness is demonstrated through experiments on two different backbone architectures, yielding compelling qualitative results that underscore its generalizability.

Weaknesses
1. The proposed function a(t) in GradSPO appears to be overly simplistic. It would be beneficial to explore alternative formulations—such as non-linear functions—or investigate whether a learnable parameterization could better capture the desired temporal behavior.
2. While Equations (10)–(12) offer a solid theoretical argument, the paper would benefit from additional empirical evidence or illustrative visualizations to more concretely demonstrate the time-step weighting bias issue inherent in SPO.
3. The experimental validation is limited in scope. It lacks evaluations on more recent and competitive backbone models such as DiT (e.g., SD3 or Flux). Moreover, it remains unclear whether the proposed method can be extended or adapted to the video generation domain, which would further demonstrate its versatility.

---

> ### Author Rebuttal · Authors · 2025-07-31
>
> Thank you for taking the time to review our paper and for offering detailed feedback. We address them below, following the order in which they appear across the review’s sections.
>
> ---
> ### **Weaknesses (W)**
> **[W-Q1] The proposed function a(t) in GradSPO appears to be overly simplistic. It would be beneficial to explore alternative formulations, such as non-linear functions.**
>
> **[W-A1]** **Table A** compares uniform timestep weighting with alternative schemes, including the widely used min-SNR weighting [1], and P2 weighting [2] for SD 1.5 on Pick-a-Pic v2. Unlike in standard diffusion training, where schemes like min-SNR or P2 often outperform uniform weighting, we find that uniform weighting performs slightly better in most metrics.
>
> While non-uniform weighting schemes [1,2] have seen wide use in diffusion model training, their use in preference learning remains limited; we find that most preference learning for text-to-image diffusion models uses uniform weighting. Although the two objectives are related, they differ in nature: standard diffusion training aims to match a target distribution, whereas diffusion preference learning focuses on maximizing the margin between winning and losing samples. This leads to different optimization dynamics, where uniform weighting may offer a better inductive bias for margin-based learning.
>
> **Table A.** GradSPO with different weighting schemes for SD 1.5 on Pick-a-Pic v2.
> |Weighting|HSPv2|Pick Score|Aesthetic Score|Image Reward|
> |--|--|--|--|--|
> |min-SNR [1]|26.72|21.26|5.6590|0.5165|
> |P2 [2]|26.79|21.25|5.6028|**0.5318**|
> |uniform|**26.86**|**21.38**|**5.7651**|0.4747|
>
> > [1] Hang et al., Efficient Diffusion Training via Min-SNR Weighting Strategy
>
> > [2] Choi et al., Perception Prioritized Training of Diffusion Models
>
>
> **[W-Q2] While Equations (10)–(12) offer a solid theoretical argument, the paper would benefit from additional empirical evidence or illustrative visualizations to more concretely demonstrate the time-step weighting bias issue inherent in SPO.**
>
> **[W-A2]** We will add the plot in the main text to provide clear empirical evidence for the timestep bias issue. To visualize its impact, we measure the difference in preference scores of $\mathbf{x}_0$ predictions at each timestep before and after preference finetuning, using noisy images from the Pick-a-Pic v2 win set (SD 1.5).
>
> We observe that in SPO, the difference in preference scores is high at early timesteps but drops off sharply in later timesteps. This exponential decrease reflects a timestep bias, where the model allocates more capacity to optimizing preference scores at earlier denoising stages, while neglecting later ones. In contrast, GradSPO exhibits a more linear decrease, indicating more balanced improvements across all timesteps, including the later ones.
>
>
>
>
> **[W-Q3] It lacks evaluations on recent backbone models.**
>
> **[W-A3]** Thank you for the suggestion. We agree that evaluating on more recent architectures would provide a stronger justification on the generality of our method. However, due to the limited time and computational constraints during the rebuttal period, we were unable to conduct additional experiments on these more resource-intensive models.
>
> That said, our experiments on SD 1.5 and SDXL, two widely adopted diffusion backbones, demonstrate consistent performance gains across multiple datasets and metrics, providing strong evidence of the effectiveness of GradSPO over prior preference learning methods.
>
> ---
> ### **Questions (Q)**
> **[Q-Q1] addressed in [W-Q1]**
>
> **[Q-Q2] Robustness checks under different settings.**
>
> **[Q-A2]** To properly analyze the performance and stability of GradSPO, we conduct ablations on key hyperparameters with all other settings fixed for SD 1.5 with results reported on Pick-a-Pic v2 (we set $\gamma=0.5$ when varying $\mu$ and $\mu=0.9$ when varying $\gamma$). Results are shown in **Table B**.
>
> We find that the method is relatively robust to variations in the EMA momentum parameter $\mu$, with only a slight drop in performance at higher values. We attribute this to an oversmoothing effect, where the EMA model lags behind the current model and adapts too slowly.
>
> In contrast, the reward guidance scale $\gamma$ has a stronger impact. Setting $\gamma$ too high increases the variance of the reward signal, especially in noisy regions, which makes convergence less stable. On the other hand, a very low $\gamma$ weakens the signal and prevents the model from learning meaningful preferences. We find $\gamma = 0.5$ to be a good balance between signal strength and noise, and use this value across both SDXL and SD 1.5 experiments.
>
> **Table B.** Ablation study on EMA momentum $\mu$ and reward guidance scale $\gamma$ for GradSPO on SD 1.5 for Pick-a-Pic v2.
> | $\gamma$ |HSPv2|Pick Score|Aesthetic Score|Image Reward|
> |-------|--------|-------------|------------------|----------------|
> | 0.25  | 26.67 | 20.71      | 5.5278           | 0.452          |
> | 0.5   | 26.86 | 21.38      | 5.7651           | 0.4747         |
> | 0.75  | 26.70  | 20.99      | 5.463            | 0.3878         |
>
> | $\mu$  | HSPv2 | Pick Score | Aesthetic Score | Image Reward |
> |--------|--------|-------------|------------------|----------------|
> | 0.9    | 26.86 | 21.38      | 5.7651           | 0.4747         |
> | 0.95   | 26.74 | 21.12      | 5.6352           | 0.4153         |
> | 0.99   | 26.68 | 21.22      | 5.7591           | 0.3871         |

---

> > ### Author Response · Authors · 2025-08-09
> >
> > Dear Reviewer XP6v,
> >
> > As the discussion phase will be ending soon, we wanted to check if you have any remaining concerns or questions. We would be happy to discuss them and provide any clarification you may need before the process concludes.
> >
> > Thank you for your time and attention.
> >
> > Best regards, Authors

---

### Official Review · Reviewer_wrXA · 2025-07-03

**Clarity:** 2
**Significance:** 4
**Originality:** 4
**Rating:** 5
**Confidence:** 5

**Summary:**

The paper introduces GradSPO. It provides an analytical study on top of stepwise preference optimization (SPO), which is a promising direction in the field, incorporating step awareness during the process.

After presenting the analysis, the paper proposes a simpler loss function compared to SPO and obtains strong results.

**Questions:**

* Could the authors show results on a recent flow-based model? I believe models like Lumina2 [1], SANA [2], etc., are small enough to conduct an experiment.
* Even though there is MaPO in the results section, I think it also deserves to be in the other relevant sections of the paper (such as Introduction).
* I found the clarity of the symbols used throughout the main text of the paper to be somewhat hard to follow. Could the authors try to be more explicit with those?
* In equation 3 and elsewhere, the authors mention the use of a timestep-dependent weighting factor. Even though it was there in DDPM [3], it's not used much in practice. Could the authors comment on that?
* The existing reward models (like PickScore [4]) are not typically timestep-aware. This work, however, relies on timestep-aware reward models. Could the authors comment on it a bit more?
* Equation 13 - can't we scale with the variance of $z$?

## References

[1] Lumina-Image 2.0: A Unified and Efficient Image Generative Framework

[2] SANA: Efficient High-Resolution Image Synthesis with Linear Diffusion Transformers

[3] Denoising Diffusion Probabilistic Models

[4] Pick-a-Pic: An Open Dataset of User Preferences for Text-to-Image Generation

**Ethical Concerns:**

["NO or VERY MINOR ethics concerns only"]

**Final Justification:**

I had asked a bunch of questions in my reviews and also raised some concerns regarding the design of experiments. The authors responded to them firmly and also validated some of the design factors with results. For example, I had asked for the effect of uniform timestep weighting, EMA, etc. They provided enough evidence to support their use. So, I will raise my score.

**Limitations:**

The authors didn't include a separate section on limitations in the main text nor did they make them clear in the main text.

**Paper Formatting Concerns:**

None noticed.

**Quality:**

3

**Strengths And Weaknesses:**

## Strengths

* Stepwise preference optimization is a promising direction in aligning diffusion models. Therefore, I believe the authors identified a good testbed to start with and trying to further reason about it and repurpose it.
* Since the method relies on a noisy approximation, the authors also introduce a noisy reduction strategy to mitigate its repercussions.
* The authors substantiate the simplification with impressive results on widely used metrics in the area.

## Weaknesses

Several important ablations and details seem missing from the main text:

  * It uses uniform timestep weighting. How about other weighting schemes?
  * What is the impact of using components like EMA and noise reduction? Even Figure 5 has win-rate comparisons, I believe a better ablation would have been to show how the loss landscape gets affected by the noise reduction strategy and also some qualitative samples.
  * For step-aware models, memory and convergence speed can be an issue. Could the authors include a discussion on this?
  * DPO and MaPO were trained on the Pick-a-Pic v2 dataset while GradSPO uses v1. Could the authors ensure that the methods being compared use the same base dataset?
  *  The effect of the stop-gradient operator in equation 10 isn't quite clear.

---

> ### Author Rebuttal · Authors · 2025-07-31
>
> Thank you for your thoughtful review. We address each of your comments below, following their order in the original review.
>
> ---
> ### **Weaknesses (W)**
> **[W-Q1] Weighting schemes other than the uniform timestep weighting**
>
> **[W-A1]** **Table A** compares uniform timestep weighting with alternative schemes, including the widely used min-SNR weighting [1], and P2 weighting [2] for SD 1.5 on Pick-a-Pic v2. Unlike in standard diffusion training, where schemes like min-SNR or P2 often outperform uniform weighting, we find that uniform weighting performs slightly better in most metrics.
>
> While non-uniform weighting schemes [1,2] have seen wide use in diffusion model training, their use in preference learning remains limited; we find that most preference learning for text-to-image diffusion models uses uniform weighting. Although the two objectives are related, they differ in nature: standard diffusion training aims to match a target distribution, whereas diffusion preference learning focuses on maximizing the margin between winning and losing samples. This leads to different optimization dynamics, where uniform weighting may offer a better inductive bias for margin-based learning.
>
> **Table A.** GradSPO with different weighting schemes for SD 1.5 on Pick-a-Pic v2.
> |Weighting|HSPv2|Pick Score|Aesthetic Score|Image Reward|
> |--|--|--|--|--|
> |min-SNR [1]|26.72|21.26|5.6590|0.5165|
> |P2 [2]|26.79|21.25|5.6028|**0.5318**|
> |uniform|**26.86**|**21.38**|**5.7651**|0.4747|
>
> > [1] Hang et al., Efficient Diffusion Training via Min-SNR Weighting Strategy
>
> > [2] Choi et al., Perception Prioritized Training of Diffusion Models
>
>
> **[W-Q2] Impact of using components like EMA and noise reduction**
>
> **[W-A2]** We will include the training curve in the main paper to illustrate the impact of EMA and noise reduction. These components lead to ~2 times faster convergence and a higher plateau in average preference score (GradSPO: ~17.8 vs. SPO: ~17.4) during training on SD 1.5. We attribute the faster convergence and higher training reward to reduced noise, which makes it easier for the diffusion model to optimize toward higher rewards.
>
>
> **[W-Q3] Issue of memory and convergence speed issue for a step-aware model**
>
> **[W-A3]** Stepwise preference methods (SPO and our GradSPO) can introduce two sources of computational overhead compared to standard Diffusion DPO [1].
>
> 1) They require a separate step-aware preference model. In our case, we reuse the publicly available pretrained checkpoint from SPO for fairness, rather than training a new model.
>
> 2) Both SPO and GradSPO perform online sampling and labeling during training using the step-aware preference model, whereas standard Diffusion DPO relies on offline-labeled data (e.g., from human or AI annotators).
>
> Despite these costs, stepwise methods converges significantly faster. Our full finetuning on SD1.5 completes in 24 hours on 4×A6000 GPUs, compared to 768 hours reported in the original Diffusion DPO paper on the same hardware.
>
> > [1] Wallace et al., Diffusion Model Alignment Using Direct Preference Optimization
>
>
> **[W-Q4] Dataset used for DPO&MaPO vs GradSPO**
>
> **[W-A4]** Pick-a-Pic v2 is an extended version of v1, preserving the same prompt quality while expanding the dataset size. Following SPO, GradSPO uses only the prompt dataset (no image supervision), selecting 4,000 prompts from Pick-a-Pic v1 and generates the images via online sampling. To ensure fairness, we compare against baselines using the same underlying prompt distribution. Although DPO and MaPO are trained on the larger v2 dataset, the prompt quality is equivalent to v1. Since our method, like SPO, depends only on prompt-level supervision, the comparison remains valid despite the size difference.
>
>
> SPO uses online sampling and relies only on the prompt dataset (no image required), selecting a subset of 4,000 prompts from Pick-a-Pic v1.
>
> For a fair comparison with SPO, we use the same prompt dataset that SPO was built on.
>
> Since our method also depends solely on a small prompt dataset, the extended size of Pick-a-Pic v2 is still a valid comparison as they share the same prompt quality.
>
>
> **[W-Q5] Effect of stop-gradient (Eq. 10) is not clear**
>
> **[W-A5]** In Eq. 10, the training objective updates the current score model $s_\theta(x_t, c, t)$ by pulling it closer to the winning score $s^w_\theta(x_t, c, t)$, and pushing it away from the losing score $s^l_\theta(x_t, c, t)$. To ensure a stable learning signal, the targets are detached using stop-gradient: $sg(s^w_{\theta_0})$ and $sg(s^l_{\theta_0})$.
>
> If the stop-gradient is not applied, the winning score would be updated during training to move toward the current score, collapsing $s^w_\theta \rightarrow s_\theta$, thereby weakening the supervision signal and destabilizing training.
>
> ---
> ### **Questions (Q)**
> **[Q-Q1] Results on flow-based model**
>
> **[Q-Q1]** Thank you for the suggestion. We agree that evaluating on recent flow-based models such as Lumina2 [1] and SANA [2] would be useful. However, due to the limited time of the rebuttal period, we were unable to conduct these new expriments on additional architectures. We note, however, that our current results, based on widely adopted benchmarks and models in diffusion preference learning literature consistently show performance improvements,  demonstrating the effectiveness of GradSPO within the established preference learning setting.
>
> > [1] Qin et al., Lumina-Image 2.0: A Unified and Efficient Image Generative Framework
>
> > [2] Xie et al., SANA 1.5: Efficient Scaling of Training-Time and Inference-Time Compute in Linear Diffusion Transformer
>
>
> **[Q-Q2] Mention of MaPO in the introduction**
>
> **[Q-Q2]** Thank you for the suggestion. We will include a mention of MaPO in the introduction in the camera-ready version.
>
> **[Q-Q3] Clarity of symbols throughout the text**
>
> **[Q-A3]** We acknowledge that some symbols, such as $x$ (image), $c$ (conditioning input, e.g., text prompt), and $T$ (total number of diffusion timesteps), are used without explicit definitions when first introduced. We agree that providing clear definitions would improve clarity for a broader audience. We will revise the paper to explicitly define all key symbols when first mentioned, especially in the Notation and Preliminaries section.
>
> **[Q-Q4] In equation 3 and elsewhere, the authors mention the use of a timestep-dependent weighting factor. Even though it was there in DDPM, it's not used much in practice. Could the authors comment on that?**
>
> **[Q-A4]** In DDPM, the timestep weighting function corresponds to the ELBO. This timestep weighting is useful when modeling likelihoods with VDM [1] and DDPM [2] achieving better likelihood estimation with an ELBO timestep weighting scheme in contrast to the uniform weighting scheme, however this weighting scheme has been shown to hurt image quality achieving lower FID on standard image benchmarks due to a timestep bias issue during training. As most image generation as well as our research is more concerned with image perceptual quality than accurate likelihood estimation, we opt for using uniform weighting instead of a timestep dependent weighting factor.
>
> > [1] Kingma et al., Variational Diffusion Models
>
> > [2] Ho et al., Denoising Diffusion Probabilistic Models
>
>
>
> **[Q-Q5] The existing reward models (like PickScore) are not typically timestep-aware. This work, however, relies on timestep-aware reward models. Could the authors comment on it a bit more?**
>
> **[Q-A5]** Our approach requires a timestep-aware reward model primarily because both SPO and GradSPO rely on *dense* supervision across intermediate denoising steps, rather than *sparse* supervision only at the final output. This is analogous to findings in LLM RL training, where dense token-level rewards have been shown to lead to better alignment than sparse response-level rewards [1,2,3].
>
> In the diffusion setting, this means scoring not just the final (clean) image but also the intermediate, noisy ones during denoising. Existing reward models like PickScore are trained only on fully denoised images and therefore produce unreliable or uninformative outputs when applied to intermediate blurry samples. To enable effective preference learning across the full denoising trajectory, we use timestep-aware reward models specifically trained on such intermediate images.
>
> > [1] Yoon et al., TLCR: Token-Level Continuous Reward for Fine-grained Reinforcement Learning from Human Feedback
>
> > [2] Zeng et al., Token-level Direct Preference Optimization
>
> > [3] Chan et al., Dense Reward for Free in Reinforcement Learning from Human Feedback
>
>
> **[Q-Q6] Equation 13 - can't we scale with the variance of z?**
>
> **[Q-A6]** While scaling with the variance of $z$ can reduce the variance of the estimator, it also unintentionally scales down its mean, $\sqrt{T} \, \beta_t \, \nabla_{x_t} r(x_t, t)$. This weakens the strength of the guidance signal. Our goal with the variance-reduced estimator in Eq. 13 is to reduce variance *without* altering the mean, so that the average reward signal strength remains consistent while the noise is suppressed. This ensures that the model receives a stable yet strong training signal. We will include this rationale in the main paper.
>
>
> **[Limitations Section Presentation]**
> We acknowledge the concern and agree that limitations should be clearly presented in the main paper. We will move the limitations section to the main paper to ensure better visibility.

---

> > ### Comment · Reviewer_wrXA · 2025-08-01
> >
> > I would like to thank the authors for providing a wonderful rebuttal. Their answers are crisp and to the point.
> >
> > I would also like to make the following suggestions:
> >
> > * It would be helpful for the end users if the paper included a direct discussion on memory consumption, based on [W-A3].
> > * Consider including most of your discussions and answers in either the main text or the appendix section of the paper. It will be beneficial for the community as notable findings.
> >
> > In light of the rebuttal, I have also raised my score to an "accept".

---

### Note · Authors · 2025-08-12

Dear AC and Reviewers,

We would like to sincerely thank you for the time, effort, and thoughtful feedback you have provided in reviewing our paper. Your comments during the rebuttal and discussion period have been invaluable in identifying areas for improvement. **Upon acceptance, we will incorporate the concerns raised as follows**:

---

## Section 1: Introduction
- Side-by-side comparison with SPO (Reviewer ccWi): We will replace *Figure 1* with a side-by-side comparison with SPO to improve clarity.

## Section 2: Notations and Preliminaries
- Notation clarity (Reviewer wrXA): We will expand the explanations for each notation in Lines 72–73 to ensure greater clarity.

## Section 3: Method
- Stop-gradient clarification (Reviewer wrXA): We will include the discussion on the stop-gradient mechanism from the rebuttal in Lines 124–125.
- Timestep bias empirical support (Reviewers wrXA, XP6v, VBZ5): We will add a new figure providing empirical evidence for the timestep bias issue, supplementing Lines 150–156.

---

## Section 4: Experiments
- Hyperparameter analysis (Reviewers XP6v, ccWi): We will extend the hyperparameter analysis in Lines 255–256.
- Fine-grained user study (Reviewer ccWi): We will replace *Figure 3* with a more detailed user study that includes “tie” results and multiple evaluation categories. We will also add a discussion of the user study pipeline and analysis in Lines 230–243.

---

## Appendix
- GenEval results (Reviewer ccWi): We will add a new Appendix J presenting GenEval results and summarizing our rebuttal and discussion on this topic.
- Failure cases (Reviewer ccWi): We will include a new Appendix K discussing the diversity issues in preference learning algorithms, as raised in the rebuttal.
- Different timestep weighting functions (Reviewers wrXA, XP6v, ccWi): We will add a new Appendix L summarizing and analyzing the different timestep weighting schemes described in the rebuttal.

---

Best regards,
*The Authors*

---

### Decision · Program_Chairs · 2025-09-17

**Decision:**

Accept (poster)

**Comment:**

The paper takes the recent SPO algorithm for Diffusion model preference fine-tuning and does a theoretical analysis in terms of a gradient guidance perspective. It then uses that framework to improve the design of the algorithm with noise reduction to prevent a temporal weighting bias. This algorithm leads to improved aesthetics and human preference alignment for image generation tasks.

Strengths: All reviewers agreed that the development of the algorithm was interesting, novel and impactful.

The bulk of complaints centered on some minor weaknesses in experiments. some ablations were suggested by reviewers and in most cases done by the authors and results shared in rebuttal. There was a concern that the gains are marginal, but I think this arises from a misunderstanding of how to interpret HPS score and other metrics. There is a concern that results are not done on the latest T2i models (like flux).  One additional concern i have is that it is unclear how important the original SPO algorithm is, since it only has 5 citations so far...so how impactful is a paper that adds analysis and improvements?

Overall however, it seems that the results justify acceptance, since it is an important problem, the derivation is non-trivial and the results are good.